# Wetlands set the pace of annual runoff in the northern Great Plains
Javad Rahmani [1], Irena F. Creed [2], Pascal Badiou[3] & Ali A. Ameli [1] ✉

Interannual variability in the runoff ratio—how much annual precipitation becomes streamflow—underpins water management, flood forecasting, and biogeochemical fluxes. In most hydrologic frameworks, this variability is attributed primarily to year-to-year climate drivers. Here, we show that in North America's wetland-rich Prairie Pothole Region, wetlands play the dominant proximate role. Using 38 years of satellite-based inundation maps and hydroclimate data from 109 catchments, we find that annual wetland inundation extent explains interannual runoff and high-flow variability more strongly than any annual or intra-annual climate index in 69% of catchments. Climate, especially snow persistence, affects wetland inundation extent, but wetland inundation exerts a stronger net control on runoff and ultimately sets the pace of annual runoff through fill–spill hydrology. Catchments exhibit wetland inundation–runoff relationships ranging from linear to strongly threshold-like, with threshold-like behavior predominant—particularly where Geographically Isolated Wetlands are abundant. These findings reveal wetlands as active regulators of ecosystem water balance and provide a landscape-explicit basis for forecasting, conservation, and adaptive water management across the region.

The Prairie Pothole Region (PPR) is a wetland-rich landscape where depressional wetlands, or "potholes", generate episodic surface connectivity that governs hydrologic and biogeochemical dynamics. The PPR—spanning ~780,000 km² of the northern Great Plains—contains five to eight million glacially carved wetlands[1,2] (Fig. 1a). Most are geographically isolated wetlands that connect intermittently to river networks via rapid overland flow, although slower groundwater exchanges can persist and help maintain wetland water levels even during dry periods[3–9]. During streamflow generation events, these wetlands toggle between "fill" phases, when water accumulates, and "spill" phases, when inundation exceeds storage thresholds and water overflows downslope[10,11]. They may also enter merge phases, when adjacent depressions coalesce to form larger wetland complexes[12]. This episodic surface connectivity not only affects flood/drought resilience[4,8], but also regulates key biogeochemical fluxes[13–16]—mediating carbon sequestration and greenhouse gas emissions, controlling downstream transfers of nitrogen and phosphorus, and determining whether wetlands act as nutrient sinks or sources[17–21]. During fill periods, potholes trap carbon and nutrients, promoting in-situ sequestration and biogeochemical removal; during spill periods, they flush them into rivers[22]. Insights into fill-spill dynamics have primarily emerged from event- and catchment-scale studies in intensively monitored catchments, with only a few regional-scale exceptions[23]. However, a generalizable understanding of

the drivers of interannual variability in hydrologic response, accounting for fill-spill dynamics, across the broader PPR remains lacking.

Prevailing hydrologic theory holds that interannual variability in catchment hydrologic response is primarily climate-driven, with landscape factors playing a secondary, moderating role. Across many settings, year-to-year fluctuations in the runoff ratio—defined as the fraction of annual precipitation exported as streamflow—are well explained by the annual aridity index (Potential Evapotranspiration (PET)/Precipitation P)[24–30]. Building on the seminal work of Milly (1994)[31], intra-annual climatic variability has been shown to further modulate interannual runoff by shaping the seasonality and timing of water and energy availability—via snow accumulation and melt timing and rainfall flashiness[32–36]. Vegetation cover and other static catchment properties (e.g., soil water storage capacity, topography, geology) have been identified as additional modulators that can adjust runoff sensitivity to climate forcing[24,25,28–30,37,38]. Together, these findings imply that although landscape features matter, annual and intra-annual climatic factors dominate interannual variability in the hydrologic response in most regions.

Against this climate-centric backdrop, the role of fill–spill dynamics in shaping interannual runoff variability across the PPR remains unresolved. While the virtual modeling experiment showed that both climate condition (dry versus wet) and wetland extent could influence annual runoff[39], it is unclear whether year-to-year changes in wetland inundation extent—

¹Department of Earth, Ocean & Atmospheric Sciences, University of British Columbia, Vancouver, BC, Canada. ²Department of Physical & Environmental Sciences, University of Toronto, Toronto, ON, Canada. ³Institute for Wetland and Waterfowl Research, Ducks Unlimited Canada, Stonewall, MB, Canada. ✉e-mail: aameli@eoas.ubc.ca

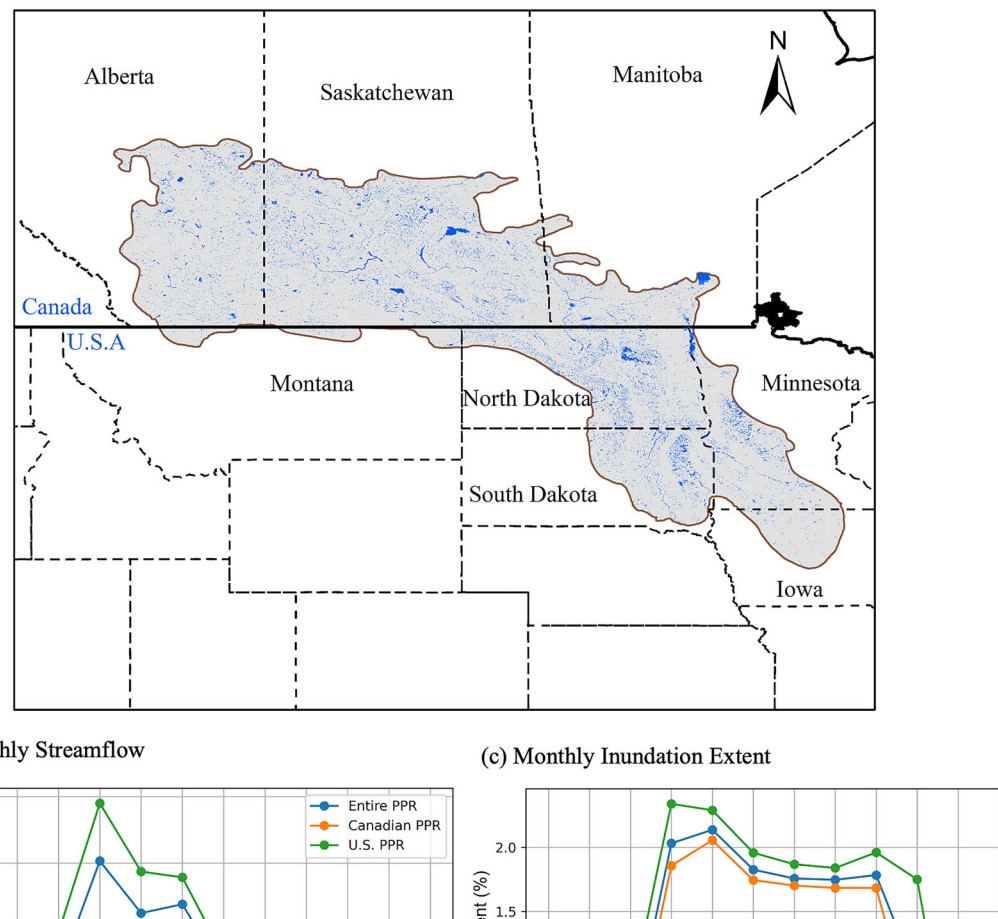

**Fig. 1 | Prairie Pothole Region (PPR) of North America, and its seasonal dynamics of streamflow and wetland inundation. a** Geographic location of the PPR, spanning parts of Canada (Alberta, Saskatchewan, Manitoba) and the United States (Montana, North Dakota, South Dakota, Minnesota, Iowa). The brown boundary delineates the region's extent across both countries. The blue pixels represent long-term inundated areas (wetlands and lakes). **b** Monthly mean river discharge (mm month⁻¹), averaged separately for 33 Canadian catchments, 76 U.S. catchments, and all 109 catchments combined. Values represent temporal averages over the full 38-year study period (1984–2021). **c** Corresponding monthly average of inundation extent (% of catchment area), spatially aggregated within Canadian and U.S. portions of the PPR and across the entire PPR region. See the Method section for details on calculating the monthly inundation extent and streamflow. All figure elements are original; the figure was generated in Python.

reflecting temporal shifts between fill- and spill-dominated states—can rival or exceed the influence of annual and intra-annual climate drivers on runoff variability. Furthermore, spatial analyses revealed a strong linear relationship between long-term contributing area (or wetland inundation extent) and long-term runoff ratio across 17 wetland-dominated catchments[40]. Whether this spatially derived linear relationship also applies to temporal (year-to-year) variations within individual catchments remains untested. In particular, modeling studies suggest that interannual runoff responses may exhibit nonlinear hysteresis, driven by dynamic feedbacks among wetland storage/inundation, connectivity, and contributing area[39,41]. This raises a fundamental question: is the relationship between annual wetland inundation extent and annual runoff ratio linear and proportional, or nonlinear—featuring thresholds, plateaus, or abrupt transitions? Identifying these functional forms is essential for determining whether wetland inundation impacts on annual runoff scale gradually or exhibit abrupt transitions in a

given catchment, with direct implications for anticipating (non)linear hydrologic and flood responses to wetland loss or climate shifts.

To date, there is no generalizable understanding of whether inter-annual runoff responses in the PPR are primarily controlled by climate, wetlands, or their interaction, nor of the functional form of the wetland inundation–runoff relationship. Both the dominant drivers and functional forms may strongly vary from catchment-to-catchment across the region. This knowledge gap has urgent practical relevance because the PPR's wetland mosaic is under mounting pressure from environmental change and management decisions[42,43]. Climate change is altering snowpack, melt timing, and rainfall intensity[44,45], while agricultural drainage continues to reduce wetland water storage capacity and, in some places, overall wetland extent[39,46,47]. It remains uncertain whether interannual hydrologic change is more strongly driven by shifting climate regimes or by loss of wetland extent due to land-use change. It is also uncertain whether these pressures push

PPR catchments from buffered, storage-regulated systems toward more linear states tightly coupled to climatic extremes[48,49].

To address these knowledge gaps, we assembled 38 years (1984–2021) of daily temperature, precipitation, and streamflow for 109 non-regulated catchments and derived two annual hydrologic response variables: (1) runoff ratio (ROR), the fraction of annual precipitation exported as streamflow, and (2) high-flow ratio (HFR), the fraction of extreme daily precipitation ($P_{95}$) converted into extreme streamflow ($Q_{95}$). We then evaluated climate drivers spanning annual and intra-annual timescales. Annual drivers included aridity index (PET/P), snow fraction, and prior-year aridity index. Intra-annual drivers included satellite-based snow persistence (SP), extreme rainfall ($R_{95}$), maximum rainfall ($R_{max}$), and monthly net water input (NWI = rainfall + snowmelt – actual evapotranspiration); both the annual maximum ($NWI_{max}$) and the April value ($NWI_{april}$) of monthly NWI were considered, which capture contributions during the key spring runoff period. We also computed the seasonality index[50] to quantify the temporal (mis)alignment between precipitation inputs and evaporative demand in each year. For each catchment and year, annual-scale values of hydrologic response variables and climatic drivers were quantified for the water year (October–September) period (See Method section).

Central to our analysis is a landscape driver that captures the inter-annual state of fill-versus-spill dominance: the annual Maximum Inundated Wetland Area (annual MIWA). We estimated annual MIWA (1984–2021) from monthly 30-m inundation maps in the global surface water dataset[51], removing pixels corresponding to lakes and rivers to isolate the extent of wetland inundation. For each catchment and year, annual MIWA is defined as the fraction of the catchment observed as water at least once during the water year (October–September). Annual MIWA thus captures the inundation extent of wetlands rather than total wetland area in each year, reflecting the landscape's hydrologically active storage capacity in a given year (see Method section). Since MIWA reflects the inundated extent each year, any reductions in wetland inundation extent—regardless of their cause—are implicitly captured in this record, thereby indirectly reflecting the potential influence of wetland loss on annual runoff.

Our first research question asked: Does annual MIWA explain more interannual variance in ROR and HFR than annual and intra-annual climatic drivers? To isolate the independent effects of wetland inundation extent and climate on annual ROR and HFR, we applied partial correlation analysis, which removes shared variance among drivers and quantifies each driver's unique explanatory power. Using partial correlation results, we classified catchments as either climate-dominated or fill–spill-dominated. A catchment was classified as climate-dominated if the partial correlation of any climatic driver (adjusted for annual MIWA) with annual ROR exceeded 0.5 and was greater than the partial correlation of annual MIWA; conversely, a catchment was classified as fill–spill-dominated if the partial correlation of annual MIWA (adjusted for each climate driver) with annual ROR exceeded 0.5 and surpassed that of all climatic drivers. These analyses jointly evaluate whether the annual water cycle in PPR catchments is governed primarily by climate-dominated processes or by fill–spill-dominated dynamics in which annual maximum wetland inundation exerts the main control on annual-scale streamflow generation (see Method section and Supplementary Note 1 for more details on assessing confounders/mediators of climate-MIWA-runoff relationships).

Our second research question asked: How does the functional form of the MIWA–runoff relationship vary across catchments? To address this, we tested whether relationships between annual MIWA and both ROR and HFR follow linear or threshold-like forms (See Method section). A linear relationship implies a consistent, proportional increase in annual runoff with greater annual wetland inundation. A threshold-like buffer form indicates that annual runoff remains minimal (i.e., fill state dominance) until a critical annual MIWA value is exceeded, after which runoff response intensifies sharply (i.e., spill state dominance). A threshold-like plateau indicates a saturating relationship, in which increases in annual MIWA beyond a threshold yield minimal changes in the annual runoff responses. We then related the functional form to the two static wetland-related

metrics derived from the 38-year record innundation: Geographically Isolated Wetlands (GIWs)' maximum inundated extent and riparian wetlands' maximum inundated extent (see Method section). These static metrics approximate the extent of wetlands in geographically isolated versus riparian areas within each catchment. In doing so, we assessed whether this spatial partitioning of wetland extent—between isolated and riparian settings—affects threshold-like versus linear functional behavior. Beyond wetland configuration, we also assessed other potential static controls on the functional form by relating it to catchment-scale topography, land-cover, soil properties, and long-term climatic conditions, allowing us to evaluate whether threshold-like versus linear behavior arises uniquely from wetland configurations or from broader landscape and climate settings.

## Results

### Temporal alignment between MIWA and Streamflow

Monthly regionally-aggregated inundation and streamflow data reveal strong temporal alignment between streamflow and inundation extent across the PPR (Fig. 1b, c). Inundation typically begins in February–March, and peaks in April in the U.S. and in April–May in Canada, reflecting the maximum extent of snowmelt. Streamflow follows the inundation pattern, peaking in April in the U.S. and in April to June in Canada, both largely driven by snowmelt. Most of the annual streamflow volume in the U.S. and Canada is generated between April and June, when inundation extent is close to its seasonal maximum. After the snowmelt period, inundation remains large—though not at its peak—through October in the U.S. and through September in Canada. This sustained inundation supports additional, though smaller, streamflow generation in response to summer and early fall rainfall: from July to October in the U.S., and from July to August in Canada. (Fig. 1).

### What drives annual runoff in the Prairie Pothole Region?

Annual runoff variability across much of the Prairie Pothole Region is governed more by annual wetland inundation extent than by climate, with most catchments exhibiting a fill–spill signal. 69% of study catchments are classified as fill–spill dominated, where annual maximum inundated wetland area (MIWA) is correlated far more strongly with interannual variability in runoff ratio (ROR) than annual or intra-annual climate drivers (Fig. 2). These catchments are concentrated in Manitoba, eastern Saskatchewan, Minnesota, North Dakota, Iowa, and western South Dakota (Fig. 3). Among all potential climatic drivers, across all study catchments, only current-year aridity index, previous-year aridity index (PY-aridity), snow persistence, and maximum monthly net water input ($NWI_{MAX}$) show notable full correlations with ROR in a portion of PPR (Fig. 2a); each weakened when adjusted for MIWA (Fig. 2b), while MIWA's partial correlation remained strong after conditioning on each climate index.

Climate influences annual runoff primarily by modulating wetland inundation extent, not by directly controlling streamflow. In ~49% (37) of fill–spill dominated catchments, snow persistence emerges as the leading climatic driver of interannual MIWA variability (median $\rho = 0.63$; Supplementary Fig. 5). Following snow persistence, current-year aridity index, PY-aridity index, and $NWI_{MAX}$ also exhibit notable correlations with MIWA across several catchments. Other climatic indices (extreme rainfall, maximum rainfall, snow fraction, seasonality index, and $NWI_{April}$) show negligible correlations with MIWA (Supplementary Fig. 3). The fill-spill dominated catchments with strong SP-MIWA full correlations also show strong full correlations between SP and ROR (median $\rho = 0.63$); however, once MIWA's effect was adjusted, the SP–ROR partial correlation declined well below the MIWA–ROR partial correlation. This indicates that the climate's effect on annual runoff operates largely through its impact on MIWA—by modulating the extent of wetland inundation, which affects catchment connectivity and runoff efficiency (See Supplementary Note 1 for detailed results on assessing confounders/mediators of climate-MIWA-runoff relationship).

Case studies further illustrate this mediation pathway (Fig. 4). Six example catchments exhibit relatively strong climate–ROR correlations,

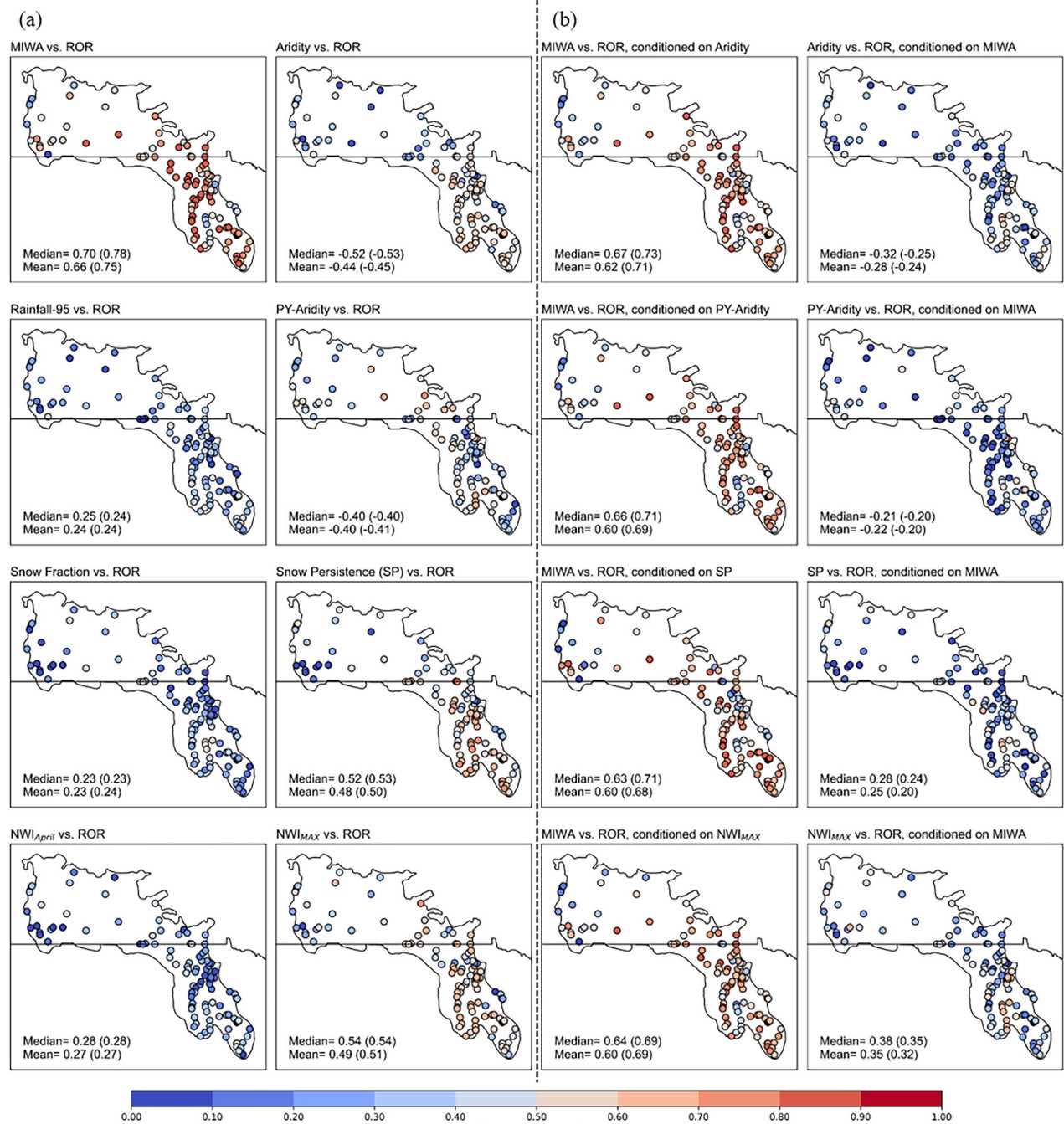

**Fig. 2 | Climatic and landscape controls on interannual runoff variability across the PPR.** Spearman correlations quantifying the strength of association between runoff ratio (ROR) and potential climatic and landscape drivers across 109 PPR catchments. The color scale reflects the absolute magnitude of correlation coefficients. Median and mean values across all 109 catchments are provided; values in parentheses denote median and mean values across the 75 fill–spill dominated catchments. **a** Full correlations between annual ROR and eight candidate drivers: maximum inundated wetland area (MIWA), current-year aridity index (Aridity), previous-year aridity index (PY-aridity), snow fraction (SF), snow persistence (SP), 95th percentile of annual rainfall (Rainfall$_{95}$), April NWI (NWI$_{April}$), and maximum monthly NWI (NWI$_{MAX}$). Here, NWI refers to Net Water Input. **b** Partial correlations isolating the independent contribution of MIWA or individual climatic variables to ROR by controlling for the alternate variable in each pair. Four partial-correlation sets are shown: First row: aridity–MIWA–ROR, second row: PY-aridity–MIWA–ROR, third row: SP–MIWA–ROR, and fourth row: NWI$_{MAX}$–MIWA–ROR. In each set, one variable is controlled while the other's association with ROR is assessed. Snow fraction, Rainfall$_{95}$, and NWI$_{April}$ were excluded from the partial correlation analysis due to their weak full correlations with ROR (see **a**). The annual seasonality index and annual maximum daily rainfall were also examined but are not shown, as their full correlations with ROR were near zero across all catchments. All figure elements are original; the figure was generated in Python.

while MIWA consistently shows stronger full and partial correlations with ROR. These six fill–spill catchments show MIWA–ROR Spearman full correlations between 0.78 and 0.89. In three catchments (05JE006, 05NG001, 05MH005), PY-aridity relates to ROR (correlations between −0.62 and −0.65; Fig. 4a–c) primarily via its association with MIWA (correlation up to −0.62; Supplementary Fig. 4a–c). In three other catchments (05053000, 05059700, 05051300), snow persistence is the best single climate driver of ROR with correlations between 0.62 and 0.69, yet MIWA

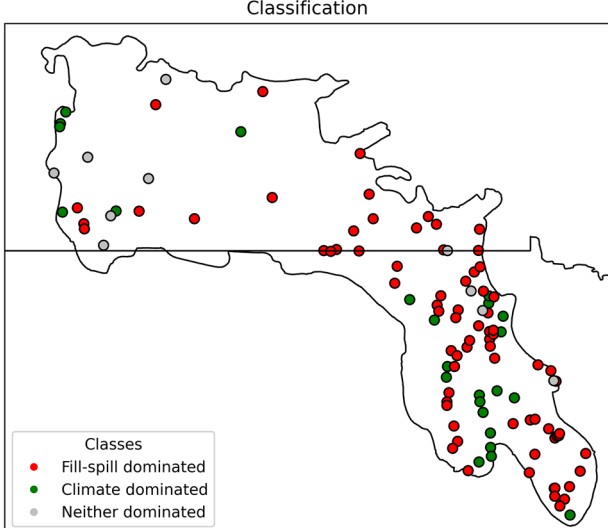

**Fig. 3 | Most study catchments exhibit fill-spill dominance.** Classification of 109 study catchments in the PPR into three categories based on the dominant driver of interannual runoff ratio (ROR) variability: fill–spill dominated (red, 69%), climate-dominated (green, 22%), and neither-dominated (gray, 9%). Categories are defined using partial correlation thresholds between ROR, maximum inundated wetland area (MIWA), and climatic indices (see Method section for classification details). All figure elements are original; the figure was generated in Python.

retains the strongest full and partial correlations with ROR (Fig. 4d–f), while SP strongly covaries with MIWA (correlation up to 0.72; Supplementary Fig. 4d–f). While climatic factors exhibit a strong correlation with MIWA along 60% (45) of fill-spill dominated catchments, 40% (30 catchments) of fill-spill dominated catchments show MIWA year-to-year variability that is not well explained by the annual and intra-annual climate indices evaluated in our paper (Supplementary Fig. 5), indicating additional controls on MIWA variability (see Discussion section).

The same wetland inundation control extends to high-flow efficiency: annual MIWA governs how readily extreme precipitation translates to high-flow conditions. In fill–spill catchments, the median full correlation between MIWA and high-flow ratio (HFR) is 0.71. This exceeds those for aridity index (−0.51), PY-aridity index (−0.32), SP (0.54), and $NWI_{MAX}$ (0.60) (Supplementary Fig. 1a), and conditioning on MIWA strongly reduces climate–HFR partial correlations (~0.05–0.46), whereas conditioning on climate leaves the MIWA–HFR partial correlations high (~0.62–0.66) (Supplementary Fig. 1b). The seasonal timing of inundation—peaking in April and remaining elevated into summer—thus sets the degree of hydrologic connectivity during the late-spring/early-summer window when snowmelt and heavy rainfall are most consequential to generate high-flow. The same six fill–spill–dominated example catchments used in the ROR variability analysis also exhibit similar behavior in the year-to-year analysis of HFR (Supplementary Fig. 2).

### What is the nature of the relation between wetland inundation extent and runoff?

In the example catchment in Saskatchewan (05JE006; Fig. 4a), annual ROR stays negligible until annual MIWA exceeds ~2% of catchment area, after which ROR increases abruptly—demonstrating how wetland networks suppress catchment-scale connectivity during fill dominance years until spill dominance years happen and the threshold of wetland inundation extent is crossed. To quantify the functional form by which annual inundation extent relates to annual-scale runoff responses in each catchment, we regressed annual MIWA on both runoff ratio (ROR) and high-flow ratio (HFR) using a power-law model across the 75 fill–spill–dominated catchments (Equation 1). The exponent $b$ of the power law model captures nonlinearity: $b > 1.25$ indicates threshold-like "buffer" dynamics (runoff

remains low until inundation crosses a tipping point, after which runoff rise sharply), $b \approx 1$ indicates an approximately linear response, and $b < 0.75$ indicates a plateau behavior in which increases in MIWA no longer enhance annual connectivity and flow beyond a threshold of wetland inundation extent (See Method Section).

Across the 75 fill-spill-dominated catchments, threshold-like buffering dominates, with 64% displaying MIWA–ROR relationships characterized by $b > 1.25$ (Fig. 5a), 8% exhibiting near-linear behavior, 3% indicating plateau-like behavior, and the remaining 25% showing weak power-law fits ($R^2 < 0.5$). These nonlinear patterns track the availability of wetland storage: buffered catchments have strongly larger potential wetland extent (median 4.25%), quantified here as the maximum inundated extent over 38-year records of inundation, and much lower long-term average ROR (median 0.04) than non-buffered catchments (1.15% and 0.17, respectively), emphasizing the role of wetland extent in damping the catchment's hydrologic response.

The strength of buffering (reflected by $b$) increases with the extent of GIWs across catchments. The nonlinearity parameter $b$ correlates most strongly with GIWs' extent (Spearman $\rho = 0.65$; Supplementary Table 1 and Fig. 5b), approximated here by GIWs' maximum inundated extent, as the percentage of catchment area, over the 38-year record. Catchments with larger wetland areas situated away from the river network exhibit stronger buffering behavior. In contrast, $b$ correlates modestly with riparian wetlands' maximum inundated extent ($\rho = 0.50$) and weakly with other climatic, geological, land-cover, and topographic attributes ($|\rho| < 0.43$; Supplementary Table 1). Consistent patterns emerge for HFR (Supplementary Fig. 6), indicating that the same GIW-driven buffering that moderates mean runoff also throttles the translation of extreme precipitation into high-flow events. This indicates that catchment configuration—not just annual climate—governs when connectivity turns on.

Landscape comparisons illustrate how modest increases in GIWs coverage can flip systems from reactive to buffered regimes. Aerial imagery for three contrasting catchments (Fig. 5c–e) shows that as GIWs extent rises from 0.73 to 3.21% of catchment area, the corresponding $b$ value climbs from 1.07 to 5.93, marking a clear shift from a precipitation-responsive, linear regime to a highly buffered system dominated by storage-mediated thresholds. This transition underscores that the additions to depressional areas can fundamentally alter flow-generation dynamics at the catchment scale.

### Discussion

Wetlands in most parts of the PPR are not passive landscape elements but active hydrologic buffers that regulate fill–spill dynamics and govern the extent and timing of catchment connectivity[52]. Annual wetland inundation, therefore, emerges as the proximate driver of both mean and high-flow responses at the annual scale. Our results demonstrate that wetland availability determines not only how much water leaves a catchment each year, but also whether that response unfolds gradually or abruptly. Many catchments exhibit threshold-like buffer behavior, in which runoff remains suppressed until inundation exceeds critical thresholds. In a few less-buffered catchments with linear relationships, streamflow generation responds more proportionally to annual climatic inputs. Our regional, observation-based findings generally align with prior catchment-scale modeling studies, which demonstrate that runoff generation in the PPR is governed by moisture-state–mediated thresholds[39,41]. Our results also suggest that the buffering effect is most pronounced in catchments with abundant geographically isolated wetlands (GIWs), which remain largely overlooked in regulatory frameworks despite their outsized role in modulating hydrologic connectivity[5,21,53]. Recognizing and preserving this buffering capacity—particularly that provided by GIWs—is central to maintaining hydrologic stability, dampening interannual variability, and mitigating flood risks in the face of ongoing climate and land-use change.

In most studied catchments, climate, especially snow persistence, modulates the year-to-year extent of wetland inundation, but wetland

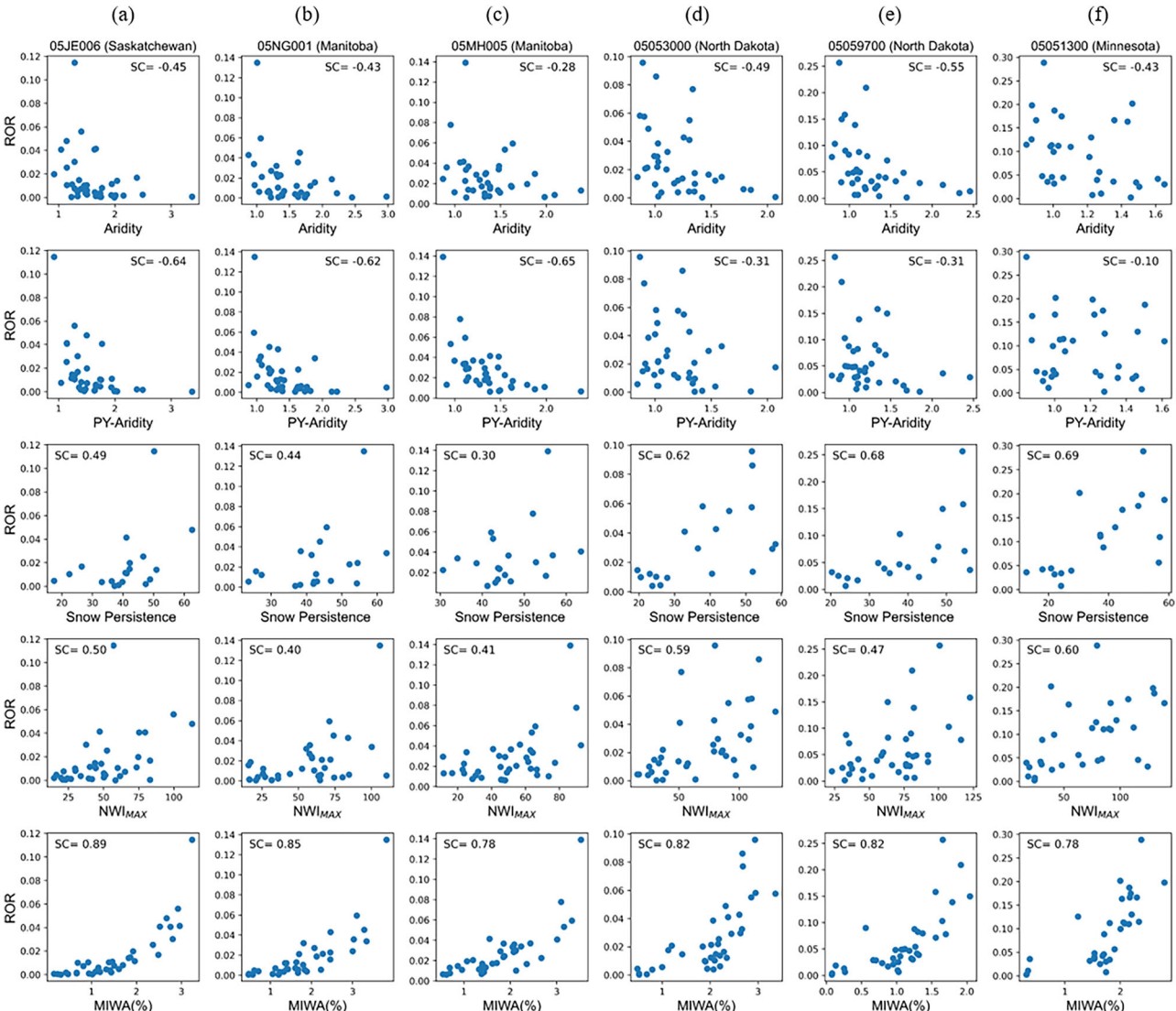

**Fig. 4 | Buffer threshold-like annual runoff responses to annual wetland inundation.** Scatter plots showing interannual relationships between aridity index (first row), PY-aridity index (second row), snow persistence (third row), NWI$_{MAX}$ (fourth row), and maximum inundated wetland area (MIWA, fifth row) with runoff ratio (ROR) across six example fill-spill dominated catchments (one catchment per column): **a** Saskatchewan (05JE006), **b** Manitoba (05NG001), **c** Manitoba (05MH005), **d** North Dakota (05053000), **e** North Dakota (05059700), **f** Minnesota (05051300).

Spearman's full correlation coefficients (SC) are reported in each panel. ROR typically increases nonlinearly with MIWA, exhibiting threshold-like behavior. These six catchments were selected because they exhibit relatively strong climate–ROR correlations, yet MIWA consistently shows stronger full and partial correlations with ROR, making them representative basins where climate influence is strong but wetland inundation extent remains the dominant driver. All figure elements are original; the figure was generated in Python.

inundation mediates these effects and ultimately sets the pace of annual runoff. In the remaining catchments, standard inter- and intra-annual climate indices fail to explain year-to-year MIWA, while MIWA still dominantly controls annual runoff. We hypothesize that in these catchments, inundation is governed by storage–release dynamics and spatially heterogeneous inputs that climate summaries miss. Subsurface pathways can subsidize specific potholes as the focus discharge from shallow aquifers and glacial stratigraphy, decoupling wetland water levels from the current-year climate and priming systems to flip once local thresholds are crossed[10,11]. Uneven inputs—including wind-redistributed snow drifts into leeward catchments and warm-season convection delivers patchy, high-intensity rainfall—amplify local inundation without altering catchment-averaged indices[54,55]. Human alteration—tile drainage[56], ditching, and small impoundments —further reorganizes wetland systems' hydrology so identical climate forcing yields different inundation states from year to year[39]. Together, these hypotheses—reserved for future investigation—may help explain the primary drivers of interannual variability in wetland

inundation in catchments where annual and intra-annual, catchment-scale climate summaries fall short.

Our findings suggest that models must be inundation-aware to perform in a threshold-prone landscape. Budyko-type[24–30] and other climate-only formulations assume proportionality and will mischaracterize catchments that respond conditionally depending on whether storage is below or above spill thresholds. Forecasting skill should improve by coupling climate forcings with dynamic indicators of inundation state, particularly within machine learning frameworks, to better predict when hydrologic connectivity activates[57]. Flood risk likewise hinges on inundation state, not climate alone. A larger extent of depressional wetlands innundation, expands lateral connections, and raises runoff efficiency. Seasonal outlooks should therefore integrate spring inundation mapping with weather forecasts to anticipate transitions from buffered to connected regimes[54,58].

Threshold-like MIWA–runoff behavior reshapes expectations for nutrient export from one of North America's most intensively farmed regions. Under buffered conditions, wetlands slow water, increase residence

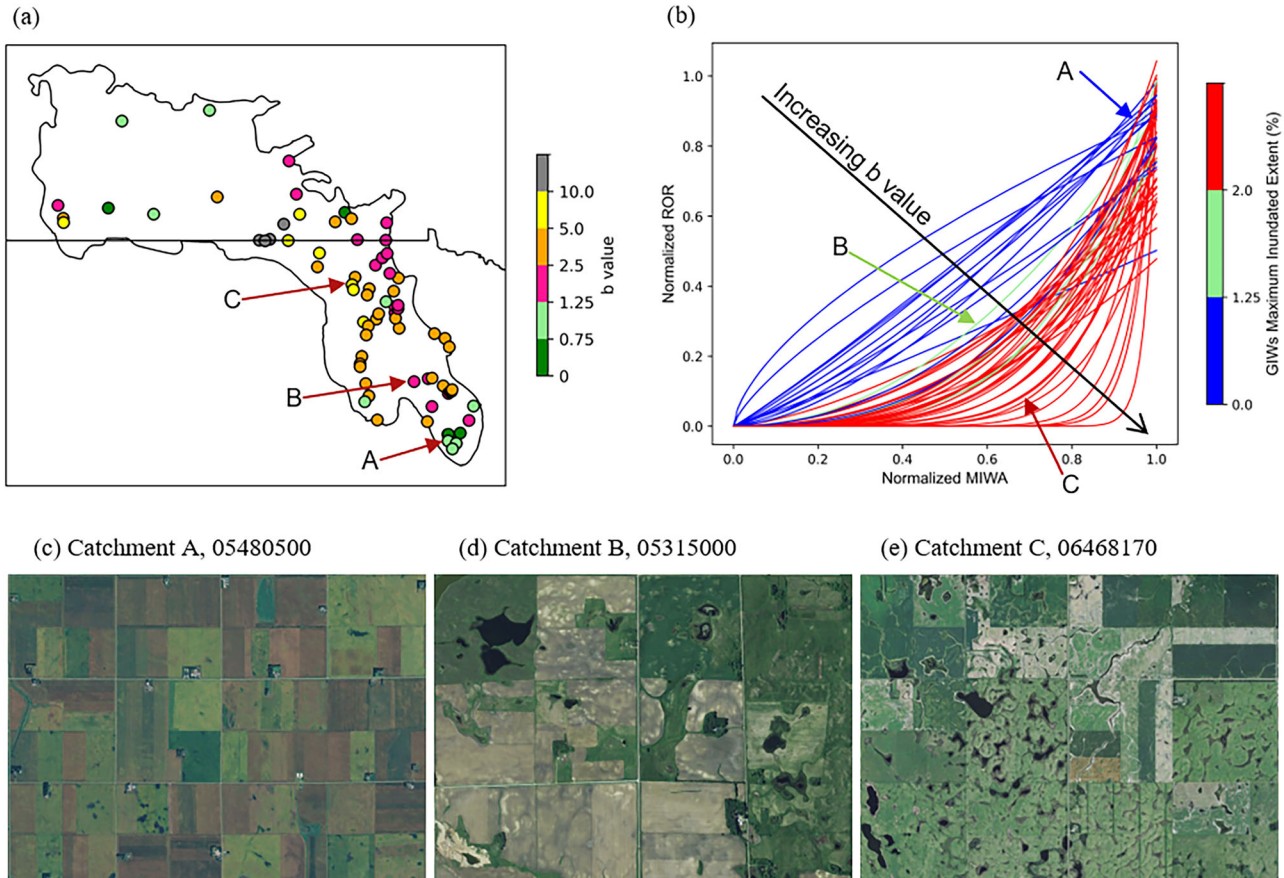

**Fig. 5 | Nonlinear runoff response to annual wetland inundation reveals widespread landscape-driven buffering across PPR catchments.** Functional characterization of the relationship between annual maximum inundated wetland area (MIWA) and annual runoff ratio (ROR) across 75 catchments where annual runoff is primarily governed by fill–spill dynamics. **a** Spatial distribution of the fitted exponent $b$ from power-law models linking MIWA to ROR, representing the strength of buffering behavior (see Method section and Equation 1). Values of $b > 1$ indicate a nonlinear "buffer" response in which annual runoff remains low unless a critical inundation threshold is exceeded, triggering rapid flow generation. Values near 1 reflect linear scaling, while $b < 1$ indicates a plateau effect, where runoff increases quickly at low MIWA and then levels off. **b** Power-law fits for 56 catchments with R² > 0.5, illustrating the shift from concave-down (plateau) to concave-up (buffer) responses as $b$ increases. Color denotes the extent of Geographically Isolated Wetlands (GIWs), as the percentage of catchment area, approximated here using GIWs' maximum inundated extent over the 38-year record of inundation. GIWs extent is strongly associated with $b$, indicating that a greater geographically isolated wetland area enhances buffering strength (see also Supplementary Table 1). Aerial imagery of three catchments with contrasting $b$ values and GIWs extents: **c** Catchment A in Iowa ($b = 1.07$, GIWs extent = 0.73%); **d** Catchment B in Minnesota ($b = 2.03$, GIWs extent = 1.38%); **e** Catchment C in North Dakota ($b = 5.93$, GIWs extent = 3.21%). **a, b** were generated in Python. **c–e**: Map data © 2024–2025 Google. Imagery ©2024–2025 Airbus.

time, and retain nutrients; once spill thresholds are exceeded, connectivity expands, and nutrient export can increase sharply. Syntheses show geographically isolated wetlands retain nitrogen and phosphorus and modulate downstream quality, highlighting the management value of maintaining depressional storage across working landscapes[20,21].

From a climate-adaptation standpoint, conserving GIWs preserves a natural stabilizer against increasingly volatile inputs. Projected declines in spring snow cover duration and intensification of warm-season storms will alter the timing and spatial structure of inputs; with intact depressional storage, many catchments should remain buffered in most years, but drainage erodes this safeguard and pushes systems toward linear, climate-dominated behavior with higher peaks and diminished drought resilience[7,39].

Finally, these results provide a mechanism-based rationale for legal recognition of GIWs as critical hydrologic infrastructure. Because their importance derives from functional buffering rather than perennial surface flow, protections tied solely to continuous surface connectivity risk dismantling the very storage that dampens floods and exports. Regulatory frameworks (e.g., interpretations of the U.S. Clean Water Act) should recognize wetlands' demonstrated role in modulating downstream

hydrology because they create episodic and threshold-dependent hydrologic connections[7,53,59].

In summary, inundation extent of wetlands, and in particular that of GIWs, in the PPR control how much water is exported each year, with inundated wetland area integrating current and antecedent climate, groundwater subsidies, storage thresholds, and spatially heterogeneous snow and rainfall, and human routing changes. An inundation-aware framing—spanning prediction, risk management, water-quality protection, climate adaptation, and conservation—offers a more faithful basis for decision-making than climate-only paradigms in this threshold-prone landscape.

## Method
### Streamflow data
We compiled daily streamflow records from all gauged stations across the PPR with data available from 1984 to 2021. These records were obtained from the United States Geological Survey and Environment and Climate Change Canada (https://wateroffice.ec.gc.ca/). We excluded stations located immediately downstream of dams or with fewer than 20 years of data meeting at least 95% annual completeness. After applying these criteria,

109 stream gauges were retained, representing our study catchments—30% located in the Canadian PPR and 70% in the U.S. PPR. These catchments span a broad range of sizes, with a median catchment area of 4197 km² and an interquartile range of 21,220 km².

## Dynamic climate data

For each study catchment, we derived a set of annual-scale climatic drivers, such as aridity index, and snow fraction (SF), as well as intra-annual climatic drivers such as snow persistence (SP), seasonality index, and monthly net water input (NWI = rainfall + snowmelt − actual evapotranspiration). To quantify these climatic indices, daily precipitation (P) and temperature data were obtained from the ERA5-Land reanalysis dataset, while daily potential evapotranspiration (PET) was sourced from the global land surface dataset. Using daily data, the annual aridity index was calculated as the ratio of total annual PET to total annual precipitation. Also, the seasonality index in each year was calculated from daily precipitation and temperature following Equation 14 of the Woods (2009)[50]. This index quantifies the alignment between precipitation and temperature, with values near −1 indicating out-of-phase conditions (precipitation peaking in winter), and values near +1 indicating in-phase conditions (precipitation primarily occurring in summer).

Monthly NWI was derived from ERA5-Land by combining monthly rainfall, snowmelt, and actual evapotranspiration. From these records, we retained both the April NWI ($NWI_{April}$) and the maximum monthly value ($NWI_{MAX}$) for each year. Snow fraction (SF) was computed as the proportion of total annual precipitation falling on days when the daily average temperature was below 0 °C[60]. We also quantified the maximum daily rainfall and the 95th percentile of rainfall in each year. Snow persistence (SP), representing the duration of seasonal snow cover, was derived from the MODIS/Terra 8-Day L3 Version 6 snow cover product (MOD10A2)[61–63]. For each 500-m grid cell, SP was quantified as the fraction of days with snow cover between January 1 and July 3, a period encompassing the typical accumulation and ablation phases across North America[58,63,64]. In each study catchment, pixel-based annual climatic indices, including SP, were aggregated to the catchment scale using the catchment's boundary polygon. To compute annual metrics from daily data, we used the water year (October–September) convention.

## Static climate (long-term average) and physical data

In addition to dynamic annual climate indices, we extracted several static catchment physio-climatic attributes to evaluate potential drivers of variation in the functional form of the wetland inundation extent–runoff (and high flow) ratios relationship (Supplementary Table 1). These attributes cover topographic, land cover, geological, and long-term climate characteristics of the study catchments. Elevation, slope, catchment boundary polygons, and catchment area were obtained from 30-m USGS digital elevation models using ArcGIS. River density was calculated as the total river length per catchment area using the dataset from Han et al.[65]. The Height Above Nearest Drainage (HAND), a proxy for hydrologic connectivity, was derived from the MERIT Hydro pixel-based dataset[66] and averaged across each catchment.

Land cover fractions (e.g., grassland, cropland, forest, urban fractions) were computed from MODIS land cover data (500-m resolution, 2001–2021), averaged, and aggregated to the catchment scale. Soil texture variables (sand, silt, and clay contents)[67] were aggregated spatially to the catchment scale. Volumetric soil moisture was obtained from ERA5-Land for four depth layers (0–7 cm, 7–28 cm, 28–100 cm, and 100–289 cm). A depth-weighted average was computed for each pixel and then aggregated to generate a catchment-level value for each study catchment. Static water table depth (500-m resolution) was derived from Janssen et al.[68] and aggregated spatially to the catchment scale. We also calculated the long-term seasonality index and long-term averages of snow fraction, aridity index, and snow persistence.

## Satellite-based wetland inundation and quantification of GIWs and Riparian wetlands maximum inundated extents

We estimated the annual extent of wetland inundation using the Landsat-based Global Surface Water dataset[51]. The 30-m resolution dataset classifies pixels as water or non-water from 1984 to 2021 using an expert system that processes imagery from Landsat 5, 7, and 8. Since the dataset captures surface inundation across all water bodies—including lakes, rivers, and wetlands—we removed pixels corresponding to mapped lakes and rivers, leaving the remaining inundation to represent the wetland inundation extent. For Canada, we used the Lakes and River dataset[69]; for the U.S., we used the National Hydrography Dataset (NHD) for rivers and the Hydro-LAKES database[70] for lakes. For each catchment and year, we calculated the annual Maximum Inundated Wetland Area (annual MIWA) as the proportion of the catchment covered by pixels classified as water (or inundated) at least once during the water year (October–September).

We also computed the maximum extent of inundation over the full 38-year record, a static metric approximating wetland extent in each catchment. This approach is consistent with previous global-scale studies that have used satellite-derived long-term inundation maps to delineate wetlands[71]. Following Cohen[5], we partitioned this long-term maximum inundated extent into two components: inundated pixels that do not intersect a 10-m buffer surrounding the mapped river network were used to quantify GIWs maximum inundated extent, while inundated pixels intersecting this buffer were used to quantify riparian wetlands' maximum inundated extent. By separating inundation in this way, these two static metrics can approximate the extent of GIWs and riparian wetlands within each catchment. The river network polyline used for this classification was sourced from Han et al.[65].

## Calculation of monthly regionally-aggregated streamflow and inundation patterns

We evaluated the seasonal alignment between wetland inundation extent and streamflow across the U.S. PPR, the Canadian PPR, and the entire PPR (Fig. 1b, c). To calculate long-term monthly streamflow for each region, we first computed the 38-year monthly average streamflow for each catchment, then averaged across all catchments within that region. The monthly average inundation extent was derived using the Global Surface Water dataset[51], accessed through the Google Earth Engine platform. For each month and country (U.S. and Canada), we calculated the 38-year average inundated area as a proportion of the study area. It is important to note that the near-zero inundation values in winter (Fig. 1c) do not indicate actual drying of wetlands, which are expected to remain water-filled but frozen. Rather, these low values reflect limitations in satellite-based inundation detection during November to late January, when extensive snow and ice cover, combined with persistent cloudiness, reduce the availability of usable Landsat imagery and hinder accurate classification of open-water surfaces.

## Relationships among climate, MIWA, and ROR

We assessed whether ROR was better explained by annual climate indices (as posited by the Budyko framework[72]), intra-annual climatic indices, or annual extent of wetland inundation, measured as annual MIWA. While the Budyko framework was originally developed to explain spatial variation in long-term water balance[73–75], it has also been widely applied, including very recently[25], to interannual analyses of ROR across diverse hydroclimatic settings[24,26–29,32,34,76,77]. To disentangle the independent influence of annual climate and MIWA on annual ROR, we employed partial correlation analysis. The same set of analysis was used to identify the independent influence of climate and MIWA on high-flow ratio (HFR), defined as the proportion of extreme precipitation ($P_{95}$) converted into extreme streamflow ($Q_{95}$), to assess whether similar relationships hold for high-flow generation dynamics. Partial Spearman correlation analysis offers a parsimonious yet sufficient approach for evaluating our central research questions across a large number of catchments. With 38 observations per catchment, a partial correlation coefficient greater than 0.5 corresponds to a p-value of approximately 0.001, indicating high statistical significance.

### Fill-spill dominated versus climate dominated catchments

In each catchment, we computed the partial correlation between annual MIWA and ROR while controlling for annual (and intra-annual) climatic indices, and vice versa. Based on these correlations, we classified catchments into three groups: (a) Fill–spill dominated: The partial correlation between annual MIWA and ROR, conditioned individually on each climatic index, is greater than 0.5 and exceeds the partial correlations between ROR and any climatic index, conditioned individually on MIWA; (b) Climate dominated: The partial correlation between ROR and at least one climatic index, conditioned on MIWA, is greater than 0.5 and larger than the partial correlation between ROR and MIWA, conditioned on that same climatic index; (c) Neither dominated: All partial correlations are less than 0.5.

### Potential confounders/mediators of climate-MIWA–ROR relationships

To evaluate whether the observed relationship between MIWA and ROR could be confounded by other climatic drivers, we tested six hypotheses using both full and partial correlation analyses. Specifically, we examined whether interannual variability in candidate climatic variables might jointly influence both MIWA and ROR, thereby creating a spurious association between them. Our evaluation framework distinguishes between confounding and mediation effects by comparing conditional and unconditional Spearman correlations. If a given climatic attribute shows strong full correlations with both MIWA and ROR, and if the partial correlation between MIWA and ROR (controlling for that climate attribute) is strongly reduced, the climate attribute is considered a likely confounder—a common driver of both variables. Conversely, if the partial correlation between MIWA and ROR (controlling for the climatic attribute) remains strong and exceeds the partial correlation between the climatic attribute and ROR (controlling for MIWA), then MIWA is interpreted as a mediator, modulating the influence of the climatic attribute on ROR. This structure supports the delineating of non-spurious associations under the assumptions of no unmeasured confounding and sufficient temporal alignment between annual MIWA and streamflow.

We tested the following six hypotheses (H) across all catchments: H1: Extreme rainfall as a shared driver: We tested if intense rainfall may simultaneously increase wetland inundation extent and ROR using the 95th percentile of daily rainfall and maximum daily rainfall (the single rainiest day) each year; H2: Snow fraction (SF) as a confounder: We tested if the proportion of annual precipitation falling as snow affects both MIWA and ROR, potentially acting as a common driver; H3: Snow persistence (SP) as a confounder: We examined if snowpack duration, reflecting accumulation and melt timing, control both spring wetland inundation and runoff events, following the prior evidence[64,78,79]; H4: Current-year aridity index as a confounder: We tested whether the annual aridity index acts as direct drivers of MIWA and ROR, or whether its apparent influence on runoff operates primarily through its effect on annual wetland inundation extent. H5: Lagged aridity index as a memory effect: We tested if antecedent climatic conditions—captured by the previous-year aridity index or the two-year average aridity index—shape current-year MIWA and ROR through legacy effects of wetness or drought; and H6: Monthly Net Water Input (NWI) as a confounder: We tested whether sub-annual (e.g., monthly) NWI controls the variabilities in both MIWA and ROR. Since runoff in the study catchments is strongly seasonal (Fig. 1b), and MIWA peaks within a narrow time window—typically in April (Fig. 1c)—seasonal metrics may capture hydrologic dynamics more effectively than annual averages. Therefore, we evaluated the extent to which interannual variations in MIWA and ROR are explained by the maximum monthly NWI ($NWI_{MAX}$) in each year or by the April NWI ($NWI_{April}$) in each year, when inundation peaks precede streamflow peaks (Fig. 1).

For each hypothesis and each catchment, we assessed both full correlations (between each climate variable and MIWA and ROR) and partial correlations (controlling for one variable while evaluating the other). This analytical framework allowed us to distinguish among confounders (shared drivers of both MIWA and ROR), mediators (indirect pathways through MIWA), and variables that play no important role. Differences in partial correlation magnitudes were key to inferring whether climate influences ROR directly or primarily through its effect on wetland inundation (See Supplementary Note 1 for details and results of these hypothesis tests).

### Functional forms of MIWA–ROR relationships

We fitted a power-law model in each fill–spill–dominated catchment, to characterize the functional form by which annual wetland inundation extent relates to annual runoff ratio:

$$ROR = a \times (MIWA)^b + c \qquad (1)$$

where *a*, *b*, and *c* are empirically estimated for each catchment. The exponent *b* quantifies the nonlinearity of the functional form: *b* < 1: Concave-down, saturating (plateau) response; *b* ≈ 1: Linear response; and *b* > 1: Concave-up, threshold-like (buffer) behavior. Analogous models were fit to MIWA-HFR interannual relationship to assess consistency across hydrologic extremes.

### Drivers of nonlinearity in MIWA–ROR relationships

To identify the key factors controlling catchment-to-catchment variation in the nonlinearity of the MIWA–ROR relationship, we computed Spearman correlations between the fitted power-law exponent *b* and the set of static physico-climatic catchment attributes summarized in Supplementary Table 1. This attribution analysis follows a similar logic to previous studies that have examined spatial variability in water balance partitioning and the functional form of the aridity–ROR relationship across catchments[77,80,81]. By correlating the degree of nonlinearity (*b*) with the catchment physiographic (e.g., GIWs maximum inundation extent), land cover, soil, and long-term climate characteristics, we assess which landscape and climate features most strongly influence whether a PPR catchment exhibits buffering, linear, or plateau behavior in linking wetland inundation to runoff.

### Reporting summary

Further information on research design is available in the Nature Portfolio Reporting Summary linked to this article.

### Data availability

All data sources used in this study are cited in the main text. The interannual and intra-annual data used to generate all figures—including MIWA, aridity index, previous year aridity index, snow persistence, snow fraction, seasonality, April NWI, maximum monthly NWI, maximum Rainfall, 95th percentile of Rainfall, 95th percentile of precipitation, 95th percentile of Streamflow, ROR, and HFR—are available at: https://doi.org/10.5281/zenodo.18153132[82]. The package includes a README that documents each dataset.

### Code availability

The code used to generate results and figures is available at: https://github.com/j-rahmani/Pothole-Inundation. The repository includes a README file with detailed instructions for installing the required libraries and running the code in Python. In addition, the expected outputs of the code—covering most of the figures used in the main manuscript and supplementary document—are provided for comparison.

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

## Acknowledgements

This research was supported by a Natural Sciences and Engineering Research Council (NSERC) Discovery Grant awarded to Ali Ameli (RGPIN-2020-04664) as well as funding from Environment and Climate Change Canada (EDF-CA-2021i023) awarded to Ali Ameli and Irena Creed. Computational support was provided by Compute Canada.

## Author contributions

J.R.: Formal Analysis, Methodology, Visualization, Investigation, Writing–Original Draft. I.F.C.: Study Design, Funding Acquisition, Writing–Review and Editing.P.B.: Writing–Review and Editing. A.A.A.: Conceptualization, Study Design, Supervision, Funding Acquisition, Writing–Original Draft.

## Competing interests

The authors declare no competing interests.
