## [Transparent Peer Review file · Communications Earth & Environment]

Wetlands Set the Pace of Annual Runoff in the Northern Great Plains

Corresponding Author: Professor Ali Ameli

This manuscript has been previously reviewed at another journal. This document only contains information relating to versions considered at Communications Earth & Environment.

Version 0:

Decision Letter:

Dear Professor Ameli,

Your manuscript titled "Wetlands Set the Pace of Annual Runoff in the Northern Great Plains" has now been seen by 2 reviewers, and we include their comments at the end of this message. They find your work of interest, but some important points are raised. We are interested in the possibility of publishing your study in Communications Earth & Environment, but would like to consider your responses to these concerns and assess a revised manuscript before we make a final decision on publication.

We therefore invite you to revise and resubmit your manuscript, along with a point-by-point response that takes into account the points raised. Please highlight all changes in the manuscript text file.

Specifically, we request that you to

****Examine the non-linear relationship between runoff ratio and inundated extent identified in your results, positioning it within the current state-of-the-art literature and comparing it with findings from catchments with similar characteristics.**

****Provide a clear justification for conducting your analysis on a calendar-year basis; if this cannot be supported, please revise the analysis using a water-year approach.**

Please submit your point-by-point responses as a separate file, distinct from your cover letter where you can add responses to the Editors' comments that you do not want to be made available to the reviewers. Word files are preferred. We recommend that any figures, tables or graphs that are included in the response to reviewers are also included in the main article or Supplementary Information.

Please use the following link to submit your revised manuscript, point-by-point response to the referees' comments (which should be in a separate document to any cover letter), a tracked-changes version of the manuscript (as a PDF file) and the completed checklist:

Link Redacted

We hope to receive your revised paper within six weeks; please let us know if you aren't able to submit it within this time so that we can discuss how best to proceed. If we don't hear from you, and the revision process takes significantly longer, we

may close your file. In this event, we will still be happy to reconsider your paper at a later date, as long as nothing similar has been accepted for publication at Communications Earth & Environment or published elsewhere in the meantime.

Please do not hesitate to contact us if you have any questions or would like to discuss these revisions further. We look forward to seeing the revised manuscript and thank you for the opportunity to review your work.

Best regards,

Leticia Santos de Lima
External Editor
Communications Earth & Environment

Nandita Basu, PhD
Associate Editor, Communications Sustainability
Consulting Editor, Communications Earth & Environment
Nature Portfolio
<https://www.nature.com/commssustain/>

EDITORIAL POLICIES AND FORMATTING

- Behavioural and social science
- Ecological, evolutionary & environmental sciences
- Life sciences

Furthermore, please align your manuscript with our format requirements, which are summarized on the following checklist: <https://www.nature.com/documents/commsj-phys-style-formatting-checklist-article.pdf> Communications Earth & Environment formatting checklist

and also in our style and formatting guide <https://www.nature.com/documents/commsj-phys-style-formatting-guide-accept.pdf> Communications Earth & Environment formatting guide .

*** DATA: Communications Earth & Environment endorses the principles of the Enabling FAIR data project (<http://www.copdess.org/enabling-fair-data-project/>). We ask authors to make the data that support their conclusions available in permanent, publically accessible data repositories. (Please contact the editor if you are unable to make your data available).

All Communications Earth & Environment manuscripts must include a section titled "Data Availability" at the end of the Methods section or main text (if no Methods). More information on this policy, is available at <http://www.nature.com/authors/policies/data/data-availability-statements-data-citations.pdf>.

If a community resource is unavailable, data can be submitted to generalist repositories such as <https://figshare.com/> or <http://datadryad.org/> Dryad Digital Repository. Please provide a unique identifier for the data (for example a DOI or a permanent URL) in the data availability statement, if possible. If the repository does not provide identifiers, we encourage authors to supply the search terms that will return the data. For data that have been obtained from publically available sources, please provide a URL and the specific data product name in the data availability statement. Data with a DOI should be further cited in the methods reference section.

REVIEWER COMMENTS:

Reviewer #1 (Remarks to the Author):

See attached review.

Reviewer #2 (Remarks to the Author):

Please see my comments in the attached file. Interesting paper.

** Visit Nature Portfolio's author and referees' website at www.nature.com/authors for information about policies, services and author benefits**

Communications Earth & Environment is committed to improving transparency in authorship. As part of our efforts in this direction, we are now requesting that all authors identified as 'corresponding author' create and link their Open Researcher and Contributor Identifier (ORCID) with their account on the Manuscript Tracking System prior to acceptance. ORCID helps the scientific community achieve unambiguous attribution of all scholarly contributions. You can create and link your ORCID from the home page of the Manuscript Tracking System by clicking on 'Modify my Springer Nature account' and following the instructions in the link below. Please also inform all co-authors that they can add their ORCIDs to their accounts and that they must do so prior to acceptance.

Version 1:

Decision Letter:

<*** REMEMBER TO ATTACH REVISIONS CHECKLIST (WORD)***>

Dear Professor Ameli,

Your manuscript titled "Wetlands Set the Pace of Annual Runoff in the Northern Great Plains" has now been seen by our reviewers, whose comments appear below. In light of their advice we are delighted to say that we are happy, in principle, to publish a suitably revised version in Communications Earth & Environment.

We therefore invite you to revise your paper one last time to address the remaining concerns of our reviewers. At the same time we ask that you edit your manuscript to comply with our format requirements and to maximise the accessibility and therefore the impact of your work.

EDITORIAL REQUESTS:

*****Please take care to match our formatting and policy requirements. We will check revised manuscript and return manuscripts that do not comply. Such requests will lead to delays. *****

SUBMISSION INFORMATION:

In order to accept your paper, we require the files listed at the end of the Editorial Requests Table; the list of required files is also available at <https://www.nature.com/documents/commsj-file-checklist.pdf> .

OPEN ACCESS:

Communications Earth & Environment is a fully open access journal. Articles are made freely accessible on publication. For further information about article processing charges, open access funding, and advice and support from Nature Portfolio, please visit <https://www.nature.com/commsenv/open-access>

Link Redacted

Best regards,

Leticia Santos de Lima
External Editor
Communications Earth & Environment

Nandita Basu, PhD
Associate Editor, Communications Sustainability
Consulting Editor, Communications Earth & Environment
Nature Portfolio
<https://www.nature.com/commssustain/>

REVIEWERS' COMMENTS:

Reviewer #1 (Remarks to the Author):

The authors have done a good job in addressing all reviewer comments. I recommend publication in current form.

Reviewer #2 (Remarks to the Author):

Thank you to the authors for their nice job addressing the reviewer comments. I do not have any further suggestions besides deleting the word "annual" on Line 40. But this seems such a minor thing given that the rest of the manuscript is ready to go. I look forward to seeing this in print.

** Visit Nature Portfolio's author and referees' website at <http://www.nature.com/authors> for information about policies, services and author benefits**

Associate Editor Evaluations:

Your manuscript titled "Wetlands Set the Pace of Annual Runoff in the Northern Great Plains" has now been seen by 2 reviewers, and we include their comments at the end of this message. They find your work of interest, but some important points are raised. We are interested in the possibility of publishing your study in *Communications Earth & Environment*, but would like to consider your responses to these concerns and assess a revised manuscript before we make a final decision on publication. We therefore invite you to revise and resubmit your manuscript, along with a point-by-point response that takes into account the points raised. Please highlight all changes in the manuscript text file.

Specifically, we request that you to

****Examine the non-linear relationship between runoff ratio and inundated extent identified in your results, positioning it within the current state-of-the-art literature and comparing it with findings from catchments with similar characteristics.**

****Provide a clear justification for conducting your analysis on a calendar-year basis; if this cannot be supported, please revise the analysis using a water-year approach.**

We thank the Associate Editor for the thoughtful evaluation and the opportunity to revise and resubmit our manuscript. We have carefully addressed all reviewer comments, with particular attention to the two major points noted above. Below, we summarize the key revisions made in this round, followed by detailed, point-by-point responses to the reviewers. Changes in the revised manuscript are highlighted in blue in the accompanying file titled "Final Manuscript_Changes Highlighted."

- **Non-linear MIWA–runoff relationship and positioning within existing literature:**

We have revised the introduction to more clearly position our study within the current state of knowledge on moisture-state–runoff relationships in wetland-dominated landscapes. In particular, we now explicitly discuss and cite recent modeling and theoretical studies—such as those by Ehsanzadeh et al., Clark and Shook, and Spence et al.—that have explored both linear and nonlinear (e.g., hysteretic or threshold-like) runoff responses to changing wetland extent. This added paragraph highlights a key gap in the literature: while spatial analyses across 17 catchments (Ehsanzadeh et al.) suggest a consistent linear relationship between long-term contributing area and runoff ratio, temporal modeling studies indicate that within-catchment responses may be far more complex. Our study directly addresses this mismatch, providing the first large-sample, observation-based assessment of how interannual variability in wetland inundation regulates annual runoff ratios across the Prairie Pothole Region. The revised paragraph added to the introduction is below for your reference:

"Furthermore, spatial analyses by Ehsanzadeh et al. (2016)³⁵ revealed a strong linear relationship between long-term contributing area (or wetland inundation extent) and long-term runoff ratio across 17 wetland-dominated catchments. Whether this spatially derived linear

relationship also applies to temporal (year-to-year) variations within individual catchments remains untested. In particular, modeling studies by Clark and Shook (2022)³⁶ and Spence et al. (2022)³⁴ suggest that interannual runoff responses may exhibit nonlinear hysteresis, driven by dynamic feedbacks among wetland storage/inundation, connectivity, and contributing area. This raises a fundamental question: is the relationship between annual wetland inundation extent and annual runoff ratio linear and proportional, or nonlinear—featuring thresholds, plateaus, or abrupt transitions? A linear response would imply simple scaling (more inundation yields more runoff), whereas a nonlinear response would resemble a buffered system with critical threshold(s) beyond which hydrologic connectivity and runoff shift rapidly. Identifying these functional forms is essential for determining whether wetland impacts on annual runoff scale gradually or exhibit abrupt transitions in a given catchment, with direct implications for anticipating (non)linear hydrologic and flood responses to wetland loss or climate shifts.”

Using the suggested references, in the discussion section, we have also put our results in the context of existing modeling literature that showed a threshold-like relationship between moisture state and runoff (as below)

“Our regional, observation-based findings generally align with prior catchment-scale modeling studies demonstrating that runoff generation in the PPR is governed by moisture-state-mediated thresholds^{34, 36}”

- **Temporal aggregation and water-year justification:**

In response to the reviewer 2’s suggestions, we have revised the analysis to use an October–September water-year framework rather than a calendar-year basis. All annual variables, correlations, and classifications have been revised accordingly, and the revised manuscript is now entirely based on the Oct–Sep water year. The resulting spatial patterns and correlation magnitudes are highly consistent with those originally presented, and the primary conclusions are unchanged—and in fact strengthened—under the water-year formulation (See Figure 1 and Table 1 below in the response to the reviewer). We have updated the Methods and Results sections to clearly document and justify this choice.

1. Reviewer #1 Evaluations:

Recommendation: Accept with minor revisions

General Comments:

This study integrates multiple spatial datasets to disentangle the relative roles of climate forcing and wetland inundation on streamflow, and at landscape scales using ca. 40 yrs of data across 109 catchments. The datasets (e.g., ranging from climate indices to satellite derived inundation to landscape attributes) and the statistical approach are impressive. The same can be said regarding the questions asked, the interpretation, and the findings, which are impactful given the documented primary controls of wetlands, particularly vulnerable isolated wetlands, on streamflow. As such, this work will likely be a substantial contribution to both the hydrologic sciences and national to global policies related to wetland conservation and mitigation. The narrative and figures are concise and effectively convey the key, and again important, contributions; with that said, I offer some suggested areas for more clarification/expansion below in detailed comments. But, in short, and given the scope and impact of this work, I recommend that it is well suited for Communications Earth and Environment.

We thank the reviewer for their thoughtful assessment of our work. We are especially grateful for the reviewer's acknowledgment of the contribution this work may make to hydrologic science and wetland conservation policy. We have carefully considered all suggested areas for clarification and expansion, and we have revised the manuscript accordingly to improve clarity, strengthen explanations, and ensure that the narrative and figures clearly convey the central conclusions. Detailed responses to the reviewer's comments are provided below.

Abstract:

As I mention in later comments, I suggest adding more about the approach that was used to understand differences in catchment-level patterns (i.e., runoff vs. MIWA) and more on the findings. That is, the work to identify catchment differences (linear vs. threshold) and attempt to understand the potential reasons why (GIWs vs. other catchment properties) could be elevated.

Also, I suggest adding a bit of text regarding the statistical approach to separate the two focal drivers; without that, readers may question the work as clearly these covary.

Great point. In response, we have revised the abstract to more clearly describe our analytical approach and findings. Specifically, we now highlight the use of partial correlation analysis to isolate the independent contributions of climate and wetland inundation despite their covariance, and we explicitly note that we quantify catchment-specific functional forms (linear, saturating, threshold-like) of the MIWA–runoff relationship and relate this variation to depressional storage characteristics, particularly the extent of geographically isolated wetlands. See two added parts below:

“Using 38 years of satellite-derived inundation maps and hydroclimate data from 109 catchments, we apply partial-correlation analysis to disentangle the influences of climate and wetland inundation extent in each catchment.”

“Furthermore, we quantify catchment-specific functional relationships between annual MIWA and runoff ratio, showing that basins span a continuum from linear to strongly threshold-like runoff-generation. Threshold-like behavior is strongly associated with Geographically Isolated Wetlands (GIWs) extents, which delay hydrologic connectivity.”

Introduction

Lines 70-72: Consider flipping the order to reflect the process...fill before spill. Also, “in-situ sequestration and removal”.

We thank the reviewer for this helpful suggestion. In response, we have revised the sentence to present the hydrologic sequence as fill before spill, which more accurately reflects process dynamics. We also incorporated the reviewer’s recommendation to reference in-situ removal alongside sequestration. The revised text now reads:

“During fill periods, potholes trap carbon and nutrients, promoting in-situ sequestration and biogeochemical removal; during spill periods, they flush them into rivers.”

Line 73: I’d be cautious to say “limited number”, and it’s not really needed here.

Great point. In response, we removed the phrase “limited number” entirely and revised the sentence to avoid specifying quantity, resulting in a more neutral and streamlined formulation. The revised sentence is:

“Insights into fill-spill dynamics have primarily emerged from event- and catchment-scale studies in intensively monitored catchments, with only a few regional-scale exceptions (e.g., Ehsanzadeh et al. (2012)¹⁹; Bacsu & Spence (2024)²⁰).”

Line 78: Depending on the audience, runoff can mean different things. I suggest defining as it’s used here explicitly.

In response, we have now explicitly defined runoff ratio at its first mention to ensure clarity for all readers. The revised text reads:

“Across many settings, year-to-year fluctuations in runoff ratio—defined as the fraction of annual precipitation exported as streamflow—”

Line 80: Here and in many other places, “reference” is used vs. the actual reference.

We thank the reviewer for noting this issue. We have replaced instances of the placeholder term “reference” with the appropriate citations throughout the manuscript.

Line 82: “broader climate seasonality” is vague here, especially given the subject is already “intra-annual climate variability”, thus the same thing.

Good point, we removed broader climate seasonality from the sentence.

Line 83: Include topography?

Good point; we added topography to the sentence

“Vegetation cover and other static catchment properties (e.g., soil water storage capacity, topography)....”

Line 84: replace “slightly” with can.

We thank the reviewer for this suggestion. We have replaced the word “slightly” with “can” in the revised manuscript to improve clarity.

Lines 87-95: Related to my comment re: abstract, this would be a good place to emphasize how catchments may exhibit different behaviors and the related knowledge gaps. As written, this paragraph and text in the following paragraph is more about knowledge gaps re: the general role of MIWA vs. climate and if the response is a threshold one, failing to address that catchments may vary in the relative roles of MIWA/climate and in their runoff response. A few added sentences could go a long way here to set the stage of the catchment-specific analyses to come.

We thank the reviewer for this helpful suggestion. We agree that the Introduction should more clearly emphasize that catchments may differ both in the relative roles of MIWA versus climate and in the form of their runoff response. In response, we have added the following paragraph to the introduction to further highlight catchment-specific behavior and related knowledge gaps.

“....., it is unclear whether year-to-year changes in wetland inundation extent—reflecting temporal shifts between fill- and spill-dominated states—can rival or exceed the influence of annual and intra-annual climate drivers on runoff variability, in a given catchment. Furthermore, spatial analyses by Ehsanzadeh et al. (2016)³⁵ revealed a strong linear relationship between long-term contributing area (or wetland inundation extent) and long-term runoff ratio across 17 wetland-dominated catchments. Whether this spatially derived linear relationship also applies to temporal (year-to-year) variations within individual catchments remains untested. In particular, modeling studies by Clark and Shook (2022)³⁶ and Spence et al. (2022)³⁴ suggest that interannual runoff responses may exhibit nonlinear hysteresis, driven by dynamic feedbacks among wetland storage/inundation, connectivity, and contributing area. This raises a fundamental question: is the relationship between annual wetland inundation extent and annual runoff ratio linear and proportional, or nonlinear—featuring thresholds, plateaus, or abrupt transitions? A linear response would imply simple scaling (more inundation yields more runoff), whereas a nonlinear response would resemble a buffered system with critical threshold(s) beyond which hydrologic connectivity and runoff shift rapidly. Identifying these functional forms is essential for determining whether wetland impacts on annual runoff scale gradually or exhibit abrupt transitions in a given catchment, with direct implications for anticipating (non)linear hydrologic and flood responses to wetland loss or climate shifts.

To date, there is no generalizable understanding of whether interannual runoff responses in the PPR are primarily controlled by climate, wetlands, or their interaction, nor of the functional form of the wetland inundation–runoff relationship, within individual catchments. Both the dominant drivers and functional forms may vary substantially across the region.”

Line 99: “drainage continuous to reduce wetland water storage capacity and ” Suggested change to reflect that wetland area can still remain but with reduced storage capacity.

We thank the reviewer for this helpful clarification. We agree that drainage can reduce wetland storage capacity even where the wetland area remains. In response, we have revised the sentence to mainly emphasize the reduction in wetland storage capacity; this reads now as:

“...while agricultural drainage continues to reduce wetland water storage capacity and, in some places, overall wetland extent.”

Lines 103-107: This state re: contribution of the work breaks up flow between the preceding state that sets up the gap (It remains uncertain) and the following text to address the gap (To address this knowledge gap).

We thank the reviewer for this helpful observation. We agree that the sentences describing the contribution of the work interrupted the flow between identifying the knowledge gap and outlining how the study addresses it. To improve clarity and narrative coherence, we have removed these sentences from the revised manuscript as suggested (~L122-130). Now, right after raising the gap, we discuss how to resolve the gap.

Line 127: “isolate depressional wetland inundation”. If I’m understanding it correctly, only the river/stream and lake features were removed, not their floodplains/riparian areas, as the latter were used in MIWA, correct? This is the correct approach in my opinion: to consider both riparian and GIWs, and then how they modulate the RO vs. MIWA response differently. But, the use of depressional wetland inundation here may suggest to some (including me) that riparian (which may not be depressional) were not included in this analysis and were removed with the streams. So, maybe a simple edit (provided that I’m correct) to say that rivers and lakes were removed to isolate wetland areas, furthering characterizing them as either riparian or GIWs.

We thank the reviewer for this careful observation and confirm that your understanding is correct: in deriving MIWA, we removed only river/stream and lake features, while retaining inundation within both riparian and geographically isolated wetlands. To avoid confusion, we removed “depressional” from “... to isolate depressional wetland inundation.”, so that the revised manuscript text is

“removing pixels corresponding to lakes and rivers to isolate the extent of wetland inundation”.

Line 128: Annual MIWA. Was the time series of MIWA examined for wetland loss trends? I see later, which I appreciate, that this analysis allowed this to be implicitly included (i.e., the effect of lower MIWA values, regardless of why) and thus to suggest the role of wetland drainage on runoff. But, and maybe not here, could there be more attention to any insights for wetland loss using the time series of MIWA, and then what that meant for runoff?

We thank the reviewer for this thoughtful comment. We did not explicitly analyze long-term wetland loss trends. However, the MIWA time series inherently captures reductions in inundated extent. Thus, any hydrologic signature of wetland drainage—manifested as persistently lower

MIWA values—is implicitly reflected in the relationship between MIWA and runoff. As wetland loss can be imprinted in the inundation record, future work could investigate these long-term signals in more detail and assess their hydrologic consequences. For the present study, however, we focus on the interannual co-variability between MIWA and runoff rather than long-term landscape change. In the revised version, we have added a clarifying sentence to emphasize the reviewer suggested point as:

“Since MIWA reflects the inundated extent each year, any reductions in wetland inundation extent—regardless of their cause—are implicitly captured in this record, thereby indirectly reflecting the potential influence of wetland drainage on annual runoff.”

Lines 150-153: Seems to repeat points from before.

We thank the reviewer for noting this redundancy. We have removed the repeated text about the implications of our works from the paragraph (~ L 175-180).

Line 159: While I understand the use of “diminishing” for saturating, its connotation is misleading a bit here, suggesting the MIWA has a diminishing effect on runoff. Rather, more increases in MIWA means increasing storage effects, correct? Consider rephrasing to emphasize that.

We thank the reviewer for this helpful clarification. We removed the word “diminishing”. It now reads as:

“A threshold-like plateau indicates a saturating relationship, in which increases in annual MIWA beyond a threshold yield minimal changes in the annual runoff responses”.

Lines 163-169: You also assessed other drivers of the exponent b ; I suggest including that here. Indeed, that was somewhat lost on me until I got to the methods, as it was only briefly mentioned in the results. There are likely opportunities to sprinkle in that you also assessed other drivers more explicitly.

We thank the reviewer for this helpful suggestion. In response, we added the following sentence to the manuscript:

“Beyond wetlands configuration, we also assessed other potential controls on the functional form by relating it to catchment-scale topography, land-cover, soil properties, and long-term climatic conditions, allowing us to evaluate whether threshold-like behavior arises uniquely from wetland geometry or from broader landscape and climate settings.”

Line 164: When I first read this, I thought that the explanatory variable was percent GIW vs. percent riparian area (not time-varying inundated area), thus a static variable for catchments unless drainage reduced wetland area. But from the methods, I see that it was the maximum inundation for each of these categories over the 38 years. That’s fine, but that could be clarified here.

This is a great point. You are correct that our analysis used the long-term maximum inundated area of geographically isolated and riparian wetlands (derived from the 38-year record). To make this explicit, we have revised the sentence to read:

“We then related the functional form to the two static inundation metrics derived from the 38-year record: Geographically Isolated Wetlands (GIWs)' maximum inundated extent and riparian wetlands' maximum inundated extent (see Method section). These static metrics approximate the extent of wetlands in geographically isolated versus riparian areas within each catchment.”

And in the method section:

“We also computed the maximum extent of inundation over the full 38-year record, a static metric approximating wetland extent in each catchment. This approach is consistent with previous global-scale studies that have used satellite-derived long-term inundation maps to delineate wetlands ⁶¹. Following Cohen (2016)⁶, we partitioned this long-term maximum inundated extent into two components: inundated pixels that do not intersect a 10-m buffer surrounding the mapped river network were used to quantify Geographically Isolated Wetlands (GIWs) maximum inundated extent, while inundated pixels intersecting this buffer were used to quantify riparian wetlands' maximum inundated extent. By separating inundation in this way, these two static metrics can approximate the extent of GIWs and riparian wetlands within each catchment.”

Lines 239-240: Consider just providing the percentages of each case.

We thank the reviewer for this suggestion. In response, we now report the percentages of catchments in each class. The text has been revised to:

“Across the 75 catchments, threshold-like buffering dominates, with 64% displaying MIWA–ROR relationships characterized by $b > 1.25$ (Fig. 5a), 8% exhibiting near-linear behavior, 3% indicating plateau-like behavior, and the remaining 25% show weak power-law fits ($R^2 < 0.5$).”

Lines 241-243: Consider bringing up this case-specific example (that leans on Fig 4a) to start this section to then transition to your approach used across all catchments to systematically assess and quantify the different possible patterns (functional forms).

Perfect suggestion; the sentence below is now moved to the beginning of the section to set the stage for our catchment-by-catchment analysis.

“In an example basin (05JE006, Saskatchewan; Fig. 4a), annual ROR stays negligible until annual MIWA exceeds ~2% of catchment area, after which ROR increases abruptly—demonstrating how depressional networks suppress catchment-scale connectivity during fill dominance years until spill dominance years happen and thresholds are crossed.”

Line 263: I believe this should be: “...responsive, linear regime to one dominated...” since the $b = 1.06$.

We thank the reviewer for this clarification. We have revised the text to read as:

“...from a responsive, linear regime to a highly buffered system dominated by storage-mediated thresholds ...”

Line 270: Consider a word other than “resistant”, as MIWA is not resistant to flows but decreases them until...

As per the suggestion of the 2nd reviewer, we merged this part into the 1st paragraph of the discussion. In that section, we used the term *attenuated* rather than *resistant*.

Line 284: The role of GIWs in regulating hydrologic connectivity is not episodic, as their presence is always regulating the degree of connectivity...their SW connectivity is just episodic. Revise.

As per the suggestion of the 2nd reviewer, we merged this part into the 1st paragraph of the discussion. The sentence was removed during merging.

Lines 297-299: Well said. Indeed, and as I mentioned before, your approach allows an assessment of how changes in MIWA (regardless of reason) affect runoff. That could maybe be emp

Great point. As noted in our response to the previous comments, we have now clarified this point in the introduction to make it more explicit.

Line 331: It's not just 'even when' but because of their disconnection and only episodic connections. That could be emphasized more.

We thank the reviewer for this helpful suggestion. We agree that the buffering role of GIWs arises not just even when they are disconnected, but because they remain disconnected for much of the year and connect only episodically when storage thresholds are exceeded. We have revised this part of the manuscript to reflect the reviewer's point.

“... downstream hydrology because they create episodic and threshold-dependent hydrologic connections”

Methods:

Line 147: How was AET derived?

Actual evapotranspiration (AET) was taken directly from the ERA5-Land reanalysis product and was used only in the calculation of monthly net water input (NWI). Specifically, for each catchment and month, we extracted ERA5-Land monthly rainfall, snowmelt, and AET, aggregated these variables to the catchment scale, and computed NWI as rainfall + snowmelt – AET. AET was not used in any other part of the analysis. We have now further clarified this point in the Method section.

“Monthly NWI was derived from ERA5-Land by combining monthly rainfall, snowmelt, and actual evapotranspiration. From these records, we retained both the April NWI (NWI_{April}) and the maximum monthly value (NWI_{MAX}) for each year.”

Lines 514-515: Same comment as before that this (“depressional wetlands and potholes”) may sound to some that you excluded riparian wetlands.

Great comment. We revised the entire manuscript text and used “wetlands” instead of “depressional wetlands and potholes” when we refer to wetlands in general.

Lines 537-543: This is results that could lead off the results section to then reference Fig 1b, c, which are main body figures but not referenced in the main body

Great suggestion. We have moved this paragraph and the following one to the beginning of the Result section in the main text, now titled “*Temporal alignment between MIWA and streamflow.*”

2. Reviewer #2 Evaluations:

In this paper, the authors determine variability in the annual runoff ratio in North America's Prairie Pothole Region is primarily a function of the moisture state of the landscape – specifically the surface water extent as measured by remote sensing. The results appear sound, though there are several unanswered questions about methodological choices that need to be clarified in the manuscript. The results align with the many previous studies that show runoff generation in this type of landscape is threshold-mediated by moisture state – which is often represented by the amount of surface water held in wetland depressions. There are several studies that show this at the catchment scale, with only a few regional scale studies. These regional studies have used observed climate and streamflow data or modelled output to glean the landscape control. The remote sensing methodology is what makes this study novel, so it is a nice complement to previous studies. Some major thoughts:

1) This is really important. Were values calculated annually or by water year? Annual values, which I think were used, will not work. Water year is appropriate because precipitation that falls the previous autumn is typically not available until spring. With some terms that are used this is not important (snow persistence), but runoff ratio is not one of them. This must be clarified and/or remedied.

We thank the reviewer for highlighting this important issue. We fully agree that carryover of autumn and early-winter snowfall into the subsequent spring runoff can be hydrologically significant in the Prairie Pothole Region, and that the choice of aggregation window may therefore influence annual runoff ratios. In response, in the revised version, we adopted an October–September water-year convention for all annual variables and have redone the full analysis on this basis.

The resulting spatial patterns and partial correlation magnitudes were highly consistent with those obtained using the original calendar-year approach (See Fig. 1 below in this R2R). Median and mean partial correlations across all climate–MIWA pairings changed only marginally, and the inferred dominance of inundation versus climate drivers remained effectively unchanged. Importantly, using the Oct–Sep water year strengthens the manuscript's primary conclusion: the number of fill–spill–dominated catchments increases from 71 (calendar year) to 75 (Oct–Sep year), while the number of neither-dominated basins decreases (See Table 1 below in R2R).

The entire revised manuscript now reflects the Oct–Sep water-year analysis, and we have added text to clarify this choice and its hydrological motivation.

Fig. 1 of R2R | Comparison of partial correlation analyses using calendar-year vs. water-year aggregation.

Spatial distribution of partial Spearman correlations between annual maximum inundated wetland area (MIWA), climatic variables, and runoff ratio (ROR) across 109 Prairie Pothole Region catchments when annual variables are aggregated using (a) the calendar year (January–December) and (b) the water year (October–September) conventions to quantify annual values. Each row corresponds to a different climate–MIWA pairing: aridity index, previous-year aridity, snow persistence (SP), and maximum monthly net water input (NWI_{MAX}). In each panel, warm colors indicate stronger positive partial correlations and cool colors indicate negative correlations. Median (and mean) correlation values across all catchments are shown below each map.

Table 1 of R2R| Comparison of catchment classifications under calendar-year and water-year aggregation.

Classification	Calendar Year	Oct-Sep Water Year
Fill-Spill Dominated	71	75
Climate Dominated	25	24
Neither Dominated	13	10

2) The non-linear relationship between runoff ratio and inundated extent illustrated in Figure 5 is intriguing as current knowledge would suggest this should be linear. This result should be put into context better by comparing and contrasting with some key missing references that would improve the manuscript. The manuscript does miss some of the literature and this has affected some of the interpretation of the data. I list those that include information that could help improve the paper in my comments below.

Thank you for suggesting these relevant references; as you will see below, we have added them and included relevant text to put our results in the context of the existing literature.

3) I am confused about the difference between MIWA and GIWs. The manuscript seems to transition to GIWs at one point and I cannot tell why. Why use MIWA for some analyses and the GIWs for others (e.g., Figures 2 and 5)? The authors say “following the classification approach of Cohen”, but that does not explain to the reader the reason for the change.

We thank the reviewer for highlighting this source of confusion. Generally, we did not want to switch between MIWA and GIWs. These two terms are different, and we used them for different analyses of our paper.

Wetlands can be classified into riparian wetlands and Geographically Isolated Wetlands (GIWs) based on their distance to the river network. Riparian wetlands are those that are directly connected to the river networks, while GIWs are defined as depressions that do not intersect a 10-m buffer around the river network (Cohen et al., 2016).

In this study, we use the term annual MIWA (annual Maximum Inundated Wetland Area), which refers to the total extent of wetland inundation (both GIWs and riparian wetland inundation) in each year. All temporal (interannual) full- and partial-correlation analyses are conducted using annual MIWA, which captures year-to-year variability in the extent of wetland inundation.

In contrast, in our cross-catchment spatial analysis, when examining the potential drivers of the nonlinearity parameter b —which is a static catchment-specific parameter—we compared the long-term (not annual) extents of geographically isolated wetlands (GIWs) versus riparian wetlands inundations, where both extents are static catchment-specific parameters and were calculated as the long-term extent of inundation within geographically isolated versus within riparian areas of each catchment.

Throughout the text, we now made significant clarifications in several places to remove this confusion (all additions were highlighted in blue)

My specific comments are below.

Line 43: Build this up. Start with “ ... overlooking the role of landscape.”

Great suggestion; it reads now as

“By overlooking the role of landscape—particularly dynamic wetland inundation—.....”

Line 44: Could read “.... Region, a landscape dominated by a 780,000 km² depressional wetland complex.”

We thank the reviewer for this helpful suggestion. For the sake of the 200-word limit in the abstract we needed to remove this sentence.

Line 52: “.... exhibit threshold-like runoff generation behaviour,”

We thank the reviewer for this helpful suggestion. We have revised the sentence accordingly.

“.....showing that basins span a continuum from linear to strongly threshold-like runoff-generation....”

Line 64: A very good reference that shows the role of groundwater fluxes in maintaining surface water storage and connection is Brannen et al. 2015 HP 29: 3862-3877.

Good suggestion. In response, we have added the Brannen et al. (2015) reference to strengthen our discussion of groundwater contributions to wetland storage and connectivity. The revised text now reads:

“...slower groundwater exchanges can persist and help maintain wetland water levels even during dry periods (Brannen et al., 2015).”

Line 65: Liebowitz has shown that the system also includes a merge function, which would be good to include in this sentence. Leibowitz et al. 2016 Wetlands 36: 323-342.

In response, we have added the merge function, following the existing structure of the sentence. The revised text is:

“.....they may also enter merge phases, when adjacent depressions coalesce to form larger wetland complexes (Leibowitz et al., 2016).”

Line 72: “event- and catchment- scale studies”

We thank the reviewer for this suggestion. We revised the sentence to refer to “event- and catchment-scale studies,” which more accurately reflects the range of work in the literature.

“Insights into fill-spill dynamics have primarily emerged from event- and catchment-scale studies in”

Line 75: It is important to recognize there have been regional scale studies that have attempted to assess wetlands' role on regional streamflow. These include:

Ehsanzadeh et al. 2012 JH 414-415: 364-373.

Bacsu and Spence 2024 CWRJ 49: 300-312

We really appreciate these suggestions. The revised text reads now as:

“Insights into fill-spill dynamics have primarily emerged from event- and catchment-scale studies in intensively monitored catchments, with only a few regional-scale exceptions (e.g., Ehsanzadeh et al. (2012)¹⁹; Bacsu & Spence (2024)²⁰)”.

Line 93: Papers that provide data and insight on the moisture state – runoff response relationship and should be cited in this paper include:

Clark and Shook 2022 WRR 58: e2022WR032694

Ehsanzadeh et al. 2015 HSJ 61: 64-78 (Figure 3)

Spence et al 2022 HESS 26: 5555-5575 (Figure 11)

We thank the reviewer for pointing us toward these important studies. Each provides valuable insight into how moisture state influences hydrologic response. In the revised manuscript, we have now added one paragraph to the introduction to acknowledge all these relevant works; see below:

“Against this climate-centric backdrop, the role of fill–spill dynamics in shaping interannual runoff variability across the Prairie Pothole Region (PPR) remains unresolved. While virtual modeling experiment by Spence et al. (2022)³⁴ showed that both climate condition (dry versus wet) and wetland extent could influence annual runoff, it is unclear whether year-to-year changes in wetland inundation extent—reflecting temporal shifts between fill- and spill-dominated states—can rival or exceed the influence of annual and intra-annual climate drivers on runoff variability, in a given catchment. Furthermore, spatial analyses by Ehsanzadeh et al. (2016)³⁵ revealed a strong linear relationship between long-term contributing area (or wetland inundation extent) and long-term runoff ratio across 17 wetland-dominated catchments. Whether this spatially derived linear relationship also applies to temporal (year-to-year) variations within individual catchments remains untested. In particular, modeling studies by Clark and Shook (2022)³⁶ and Spence et al. (2022)³⁴ suggest that interannual runoff responses may exhibit nonlinear hysteresis, driven by dynamic feedbacks among wetland storage/inundation, connectivity, and contributing area.”

Using the suggested references, in the discussion section, we have also put our results in the context of existing modeling literature that showed a threshold-like relationship between moisture state and runoff (as below)

“Our regional, observation-based findings generally align with prior catchment-scale modeling studies demonstrating that runoff generation in the PPR is governed by moisture-state–mediated thresholds^{34, 36}”

Line 103: Spence's 2022 paper listed above used hydrological modelling to disentangle the climate and wetland drainage signals and should be cited here. Ehsanzadeh's two papers also did.

Great point. We added the part below to the introduction to reflect this

“Against this climate-centric backdrop, the role of fill–spill dynamics in shaping interannual runoff variability across the Prairie Pothole Region (PPR) remains unresolved. While virtual modeling experiment by Spence et al. (2022)³⁴ showed that both climate condition (dry versus wet) and wetland extent could influence annual runoff.....”

Line 106: “Overlooked” might be a strong word here. The Canadian and American Midwest water resources communities are well aware of this phenomenon. I agree that Europeans don't understand at all.

We thank the reviewer for this observation. In response to feedback from the first reviewer, we have removed this sentence entirely from the revised manuscript.

Line 111 and throughout: The symbol nomenclature is really clunky and inconsistent through the paper. Why not use Q/P instead of ROR? Or Q95/P95 instead of HFR? S/P instead of SF? The former forms are more transparent. The aridity index is first introduced as PET/P, but then as “Aridity” afterwards. I suggest summarizing these in a table to start. That could allow you to delete much of what is in the paragraph from Lines 108-122.

We thank the reviewer for this thoughtful suggestion. We agree that alternative expressions such as Q/P, Q95/P95, and S/P are intuitive and transparent. However, the current variable names (e.g., ROR, SF) have been widely used in previous studies on runoff ratio and rainfall-runoff relationships. Retaining this notation helps ensure continuity with the existing literature. Additionally, displaying fractional forms (e.g., Q/P) throughout the main text could detract from the manuscript's visual clarity and readability. Regarding the suggestion to include a table of indices: given the format of this letter-type manuscript (and formal and informal limitations on the number of figures/tables), we felt it would not be optimal to devote a full table solely to index definitions. Nonetheless, we have carefully reviewed the manuscript to ensure all notations are used consistently—for example, expressing the aridity index in a uniform manner throughout—and have clarified key terms where appropriate. We have also used expanded versions of important abbreviations, such as snow persistence throughout the revised manuscript.

Line 148: Sorry the confusion. Do you mean “annual maximum” when you say “seasonal”?

We thank the reviewer for pointing out this ambiguity. Yes, we intended “seasonal” to refer to the annual maximum inundated extent. To prevent confusion, we have revised the text to explicitly state “annual maximum” in place of “seasonal.”

“.....by fill–spill-dominated dynamics in which annual maximum wetland inundation exerts”

Line 150: Ehsanzadeh et al 2015 and Spence et al 2022 arguably have systematically assessed interannual timescales with their work evaluating wetland influence on flood frequency.

We now acknowledge these contributions in the revised manuscript. Please see previous responses.

Line 203: Please explain why these six?

Good point. The six catchments were selected because they all exhibit relatively strong climate–ROR correlations, yet in every case the MIWA–ROR (partial) correlations remain stronger. These examples therefore highlight an important pattern in our dataset: even in catchments where climate appears to strongly influence annual runoff, dynamic wetland inundation still provides stronger explanatory signal. We selected these catchments to demonstrate that MIWA meaningfully contributes to interannual runoff variability even under conditions where climate effects are substantial. A brief clarification of this rationale has been added to the manuscript. We have added the following sentence to figure 4 caption:

“Six example catchments were selected because they exhibit relatively strong climate–ROR correlations, while MIWA consistently shows stronger full and partial correlations with ROR, making them representative basins where climate influence is substantial, but wetland inundation remains dominant.”

Line 252: Where are the data that show catchments with larger areas of wetlands located away from the network show greater buffer capacity? I am sorry if I missed it. Referencing a figure would be adequate.

We thank the reviewer for raising this point. GIWs are, by definition, wetlands located away from the river network, whereas riparian wetlands lie adjacent to and connected with the channel system. Our spatial correlation analysis shows that the catchments with larger geographically isolated inundation extent exhibit larger nonlinearity parameter b of the MIWA–ROR relationship. This indicates that catchments with larger areas of wetlands located away from the river network (i.e., larger GIW extent) exhibit greater buffering capacity and stronger threshold-like behavior (Figure 5 and Table 1 in supplementary material).

Line 263: This sentence belongs in the Discussion section. Also, it would be good discuss the results in the context of Shook’s work on wetland influence on hydrological connectivity. Shook et al. 2021 JH 593: 125846 Shook and Pomeroy 2011 HP 25: 3890-3898.

Thank you for the thoughtful suggestion. As recommended, we have merged this part into the opening paragraph of the discussion section. We also added Shook’s work (Shook et al. 2021 JH 593), see below

“Wetlands in most parts of the PPR are not passive landscape elements but active hydrologic buffers that regulate fill–spill dynamics and govern the extent and timing of catchment connectivity”⁷²

Line 267: This paragraph belongs in the Discussion section.

As recommended, we have merged this part into the opening paragraph of the discussion section.

Line 304-307: This is nice; excellent points to make.

We thank the reviewer for this positive feedback.

Line 334: The data presented here do not support an evaluation of the hydrograph, just the annual scale response (Q/P). Perhaps rephrase to: "... control how much water is exported each year, with MIWA integrating"

We agree that our analysis is limited to annual-scale responses (Q/P) and does not evaluate the full hydrograph. To avoid overstating the scope of our results, we have revised the sentence, as suggested, to:

"...control how much water is exported each year, with MIWA integrating"

Line 333-339: While I agree with this statement, this study did not evaluate thresholds, hysteresis or groundwater subsidies, or anything else in this list, and how it manifests in inundated area. This final section needs refocusing to what this study found.

Thank you for this helpful comment. We agree that the original paragraph extended beyond the direct scope of our findings. In the revised version, we have refocused this section. Specifically, we now frame potential explanations—such as groundwater subsidies and human alteration—as hypotheses to be explored in future research, rather than as conclusions drawn from our data. We have also removed references to thresholds and hysteresis to avoid overinterpretation. The revised paragraph now reads as follows

"In most studied catchments, climate, especially snow persistence, modulates the year-to-year extent of wetland inundation, but wetland inundation mediates these effects and ultimately sets the pace of annual runoff. In the remaining catchments, standard inter- and intra-annual climate indices fail to explain year-to-year MIWA, while MIWA still dominantly controls annual runoff. We hypothesize that in these catchments, inundation is governed by storage–release dynamics and spatially heterogeneous inputs that climate summaries miss. Subsurface pathways can subsidize specific potholes as shallow aquifers and glacial stratigraphy focus discharge, decoupling wetland water levels from the current-year climate and priming systems to flip once local thresholds are crossed^{10,11}. Uneven inputs—including wind-redistributed snow drifts into leeward basins and warm-season convection delivers patchy, high-intensity rainfall—amplify local inundation without altering basin-mean indices^{45,46}. Human alteration—drainage, ditching, and small impoundments—further reorganizes wetland systems' hydrology so identical climate forcing yields different inundation states from year to year³⁴. Together, these hypotheses—reserved for future investigation—may help explain the primary drivers of interannual variability in MIWA in catchments where annual, catchment-scale climate summaries fall short"

We include this revised paragraph to offer context for an important finding: in some catchments, our climatic indices did not explain the year-to-year variability in wetland inundation. Rather than leaving this unexplained, we suggest plausible hypotheses and explicitly encourage future work to evaluate them. We believe this addition is important to guide follow-up research and help readers understand why climate–inundation linkages may not always be straightforward.

Figure 1: Something is wrong with the data in panel c). It implies the wetlands dry out each winter, when in all likelihood they just freeze.

We thank the reviewer for this insightful observation. We agree that near-zero inundation values in winter do not reflect actual drying of wetlands, which likely remain water-filled but frozen during this period. These low values stem from limitations in satellite-based inundation detection between November and late January: extensive snow cover, surface ice, and persistent cloudiness reduce both the availability of usable Landsat imagery and the accuracy of open-water classification. As such, winter inundation is underestimated—not physically absent. This limitation has minimal impact on our analysis, as streamflow is negligible during these months and any additional inundation is unlikely to exceed what is already captured in adjacent seasons.

To acknowledge this limitation, we added the following part to the paper:

“It is important to note that the near-zero inundation values in winter (Fig. 1c) do not indicate actual drying of wetlands, which are expected to remain water-filled but frozen. Rather, these low values reflect limitations in satellite-based inundation detection during November to late January, when extensive snow and ice cover, combined with persistent cloudiness, reduce the availability of usable Landsat imagery and hinder accurate classification of open-water surfaces”

Figure 5: Nice figure. The last sentence of the caption needs rephrasing because it is panel b that illustrates how coverage of wetlands enhances runoff buffering. The pictures just visualize it.

We thank the reviewer for this helpful clarification. We removed the entire sentence from the end of the caption of Figure 5.

Line 449: More details on why stations were selected is needed. I assume the “exotic” rivers were excluded. What other criteria were used to select stations.

We thank the reviewer for this comment. In large-sample hydrology studies, the aim is typically to include as many catchments as possible while applying only essential exclusion criteria. Consistent with this approach, we began with all gauging stations within the Prairie Pothole Region that had available streamflow observations. We then applied a small set of fundamental quality filters: stations were retained if they (i) had at least 20 years of data with a minimum of 95% annual completeness, and (ii) were not located immediately downstream of dams or major flow regulation structures. Beyond these basic exclusions, all remaining stations were included to ensure broad spatial coverage across the region. We have to acknowledge that our selected categories cover a wide range of sizes (i.e. catchment area) and flow regimes. We

already have the below description in the method section

“We compiled daily streamflow records from all gauged stations across the PPR with data available from 1984 to 2021. These records were obtained from the United States Geological Survey and Environment and Climate Change Canada (<https://wateroffice.ec.gc.ca/>). We excluded stations located immediately downstream of dams or with fewer than 20 years of data meeting at least 95% annual completeness. After applying these criteria, 109 stream gauges were retained, representing our study catchments—30% located in the Canadian PPR and 70% in the U.S. PPR. These catchments span a broad range of sizes, with a median catchment area of 4,197 km² and an interquartile range of 21,220 km². ”

Line 461: Was snowmelt also taken from ERA5?

Yes, monthly snowmelt was taken directly from the ERA5-Land reanalysis product to ensure source-consistency of all components (i.e. rainfall, snowmelt, and actual evapotranspiration) of Net Water Input NWI. Note that Snow Persistence was alternatively calculated using available satellite data.

Line 483: See my comment about Figure 1 in regards to minima of inundation. This needs to be clear.

We thank the reviewer for raising this point. As noted in our response to the Figure 1 comment, winter inundation minima in the Landsat-based dataset reflect remote-sensing limitations (snow/ice cover and cloudiness) rather than true physical drying of wetlands. In the revised manuscript, we have clarified this explicitly as:

“It is important to note that the near-zero inundation values in winter (Fig. 1c) do not indicate actual drying of wetlands, which are expected to remain water-filled but frozen. Rather, these low values reflect limitations in satellite-based inundation detection during November to late January, when extensive snow and ice cover, combined with persistent cloudiness, reduce the availability of usable Landsat imagery and hinder accurate classification of open-water surfaces. ”

In addition, in the revised analysis, all annual variables are now computed using the Oct–Sep water year rather than the calendar year. This change ensures that our annual aggregation aligns with the region's hydrologic cycle and avoids misinterpretation of winter “minimal” inundation values. We have updated the text accordingly to make this clear in the Methods and Results.

Line 487 – 507: I am not sure where these data were used.

We thank the reviewer for this comment. The static catchment features listed in Lines 487–507 of the original manuscript were used as candidate drivers of the MIWA–ROR nonlinearity parameter b in the subsection “Drivers of nonlinearity in MIWA–ROR relationships” (the final subsection of the Methods). In that analysis, we computed spatial correlations between b and each static attribute—including topography, soils, land cover, long-term climate, and etc—to assess which landscape characteristics best explain variation in functional form across catchments.

Line 518: I understand why the authors would like to use “wetland” but the Pekel dataset does not measure wetlands, but inundated area. A qualifying statement (I am sorry if I missed it) would be good to include.

This is an excellent point. The Pekel dataset detects surface inundation from all water bodies—including lakes, rivers, and wetlands—not wetlands per se. In our workflow, we first extracted inundation from the Pekel dataset and then removed mapped lakes and rivers. The remaining inundated pixels—those not associated with river or lake polygons—were treated as wetland-relevant inundation for the purpose of deriving the annual Maximum Inundated Wetland Area (MIWA). Importantly, annual MIWA does not reflect the total extent of wetlands in each catchment and each year, but rather the extent of “wetland inundation” in a given year—reflecting the landscape's hydrologically active storage in each year. In this sense, MIWA serves as an indicator of catchment moisture state, capturing how much of the wetland storage network is activated in each year rather than delineating wetland boundaries. We have revised the text to clarify this distinction in the method section:

“We estimated the annual extent of wetland inundation using the Landsat-based Global Surface Water dataset developed by Pekel et al. (2016)⁴³. The 30-m resolution dataset classifies pixels as water or non-water from 1984 to 2021 using an expert system that processes imagery from Landsat 5, 7, and 8. Since the dataset captures surface inundation across all water bodies—including lakes, rivers, and wetlands—we removed pixels corresponding to mapped lakes and rivers, leaving the remaining inundation to represent the wetland inundation extent. For Canada, we used the Lakes and River dataset ⁵⁹; for the U.S., we used the National Hydrography Dataset (NHD) for rivers and the HydroLAKES database ⁶⁰ for lakes. For each catchment and year, we calculated the annual Maximum Inundated Wetland Area (annual MIWA) as the proportion of the catchment covered by pixels classified as water (or inundated) at least once during the water year (October–September)”

And then in the main text:

“Central to our analysis is a landscape driver that captures the interannual state of fill-versus-spill dominance: the annual Maximum Inundated Wetland Area (annual MIWA). We estimated annual MIWA (1984–2021) from monthly 30-m inundation maps in the global surface water dataset developed by Pekel et al. (2016)⁴³, removing pixels corresponding to lakes and rivers to isolate the extent of wetland inundation. For each catchment and year, annual MIWA is defined as the fraction of the catchment observed as water at least once during the water year (October–September). Annual MIWA thus captures the inundation extent of wetlands—the dominant storage features in PPR catchments—rather than total wetland area in each year, reflecting the landscape's hydrologically active storage capacity in a given year (See Data section for more details).”

Line 524: Add an acronym here for the geographically isolated wetlands (GIWs).

We thank the reviewer for this suggestion. We have added the acronym GIWs at its first mention to improve clarity and consistency throughout the manuscript.

Lines 537-549: This content reads like Results can could be moved

Thank you for this suggestion. We have moved these two paragraphs into the Results section as the first results subsection, now titled “Temporal alignment between MIWA and streamflow.”

REVIEWERS' COMMENTS:

Reviewer #1 (Remarks to the Author):

The authors have done a good job in addressing all reviewer comments. I recommend publication in current form.

Reviewer #2 (Remarks to the Author):

Thank you to the authors for their nice job addressing the reviewer comments. I do not have any further suggestions besides deleting the word "annual" on Line 40. But this seems such a minor thing given that the rest of the manuscript is ready to go. I look forward to seeing this in print.

We thank both reviewers for their constructive feedback throughout the review process. In response, we have removed the word “annual” from the sentence in the abstract. Additionally, we made minor revisions to the abstract, main text, and figure captions to ensure alignment with the Nature Portfolio formatting requirements outlined in the document “Editorial_Requests_Table_1768886739_49.”

In this paper, the authors determine variability in the annual runoff ratio in North America's Prairie Pothole Region is primarily a function of the moisture state of the landscape – specifically the surface water extent as measured by remote sensing. The results appear sound, though there are several unanswered questions about methodological choices that need to be clarified in the manuscript. The results align with the many previous studies that show runoff generation in this type of landscape is threshold-mediated by moisture state – which is often represented by the amount of surface water held in wetland depressions. There are several studies that show this at the catchment scale, with only a few regional scale studies. These regional studies have used observed climate and streamflow data or modelled output to glean the landscape control. The remote sensing methodology is what makes this study novel, so it is a nice complement to previous studies. Some major thoughts:

- 1) This is really important. Were values calculated annually or by water year? Annual values, which I think were used, will not work. Water year is appropriate because precipitation that falls the previous autumn is typically not available until spring. With some terms that are used this is not important (snow persistence), but runoff ratio is not one of them. This must be clarified and/or remedied.
- 2) The non-linear relationship between runoff ratio and inundated extent illustrated in Figure 5 is intriguing as current knowledge would suggest this should be linear. This result should be put into context better by comparing and contrasting with some key missing references that would improve the manuscript. The manuscript does miss some of the literature and this has affected some of the interpretation of the data. I list those that include information that could help improve the paper in my comments below.
- 3) I am confused about the difference between MIWA and GIWs. The manuscript seems to transition to GIWs at one point and I cannot tell why. Why use MIWA for some analyses and the GIWs for others (e.g., Figures 2 and 5)? The authors say “following the classification approach of Cohen”, but that does not explain to the reader the reason for the change.

My specific comments are below.

Line 43: Build this up. Start with “ ... overlooking the role of landscape.”.

Line 44: Could read “.... Region, a landscape dominated by a 780,000 km² depressional wetland complex.”

Line 52: “.... exhibit threshold-like runoff generation behaviour,”

Line 64: A very good reference that shows the role of groundwater fluxes in maintaining surface water storage and connection is Brannen et al. 2015 HP 29: 3862-3877.

Line 65: Liebowitz has shown that the system also includes a merge function, which would be good to include in this sentence.

Leibowitz et al. 2016 Wetlands 36: 323-342.

Line 72: “event- and catchment- scale studies”

Line 75: It is important to recognize there have been regional scale studies that have attempted to assess wetlands’ role on regional streamflow. These include:

Ehsanzadeh et al. 2012 JH 414-415: 364-373.

Bacsu and Spence 2024 CWRJ 49: 300-312

Line 93: Papers that provide data and insight on the moisture state – runoff response relationship and should be cited in this paper include:

Clark and Shook 2022 WRR 58: e2022WR032694

Ehsanzadeh et al. 2015 HSJ 61: 64-78 (Figure 3)

Spence et al 2022 HESS 26: 5555-5575 (Figure 11)

Line 103: Spence’s 2022 paper listed above used hydrological modelling to disentangle the climate and wetland drainage signals and should be cited here. Ehsanzadeh’s two papers also did.

Line 106: “Overlooked” might be a strong word here. The Canadian and American Midwest water resources communities are well aware of this phenomenon. I agree that Europeans don’t understand at all.

Line 111 and throughout: The symbol nomenclature is really clunky and inconsistent through the paper. Why not use Q/P instead of ROR? Or Q₉₅/P₉₅ instead of HFR? S/P instead of SF? The former forms are more transparent. The aridity index is first introduced as PET/P, but then as “Aridity” afterwards. I suggest summarizing these in a table to start. That could allow you to delete much of what is in the paragraph from Lines 108-122.

Line 148: Sorry the confusion. Do you mean “annual maximum” when you say “seasonal”?

Line 150: Ehsanzadeh et al 2015 and Spence et al 2022 arguably have systematically assessed interannual timescales with their work evaluating wetland influence on flood frequency.

Line 203: Please explain why these six?

Line 252: Where are the data that show catchments with larger areas of wetlands located away from the network show greater buffer capacity? I am sorry if I missed it. Referencing a figure would be adequate.

Line 263: This sentence belongs in the Discussion section. Also, it would be good discuss the results in the context of Shook’s work on wetland influence on hydrological connectivity.

Shook et al. 2021 JH 593: 125846

Shook and Pomeroy 2011 HP 25: 3890-3898.

Line 267: This paragraph belongs in the Discussion section.

Line 304-307: This is nice; excellent points to make.

Line 334: The data presented here do not support an evaluation of the hydrograph, just the annual scale response (Q/P). Perhaps rephrase to: "... control how much water is exported each year, with MIWA integrating"

Line 333-339: While I agree with this statement, this study did not evaluate thresholds, hysteresis or groundwater subsidies, or anything else in this list, and how it manifests in inundated area. This final section needs refocusing to what this study found.

Figure 1: Something is wrong with the data in panel c). It implies the wetlands dry out each winter, when in all likelihood they just freeze.

Figure 5: Nice figure. The last sentence of the caption needs rephrasing because it is panel b that illustrates how coverage of wetlands enhances runoff buffering. The pictures just visualize it.

Line 449: More details on why stations were selected is needed. I assume the "exotic" rivers were excluded. What other criteria were used to select stations.

Line 461: Was snowmelt also taken from ERA5?

Line 483: See my comment about Figure 1 in regards to minima of inundation. This needs to be clear.

Line 487 – 507: I am not sure where these data were used.

Line 518: I understand why the authors would like to use "wetland" but the Pekel dataset does not measure wetlands, but inundated area. A qualifying statement (I am sorry if I missed it) would be good to include.

Line 524: Add an acronym here for the geographically isolated wetlands (GIWs).

Lines 537-549: This content reads like Results can could be moved.

MS # COMMSENV-25-4669-T

Title: *Wetlands set the pace of annual runoff in the Northern Great Plains*

Recommendation: **Accept with minor revisions**

General Comments: This study integrates multiple spatial datasets to disentangle the relative roles of climate forcing and wetland inundation on streamflow, and at landscape scales using ca. 40 yrs of data across 109 catchments. The datasets (e.g., ranging from climate indices to satellite derived inundation to landscape attributes) and the statistical approach are impressive. The same can be said regarding the questions asked, the interpretation, and the findings, which are impactful given the documented primary controls of wetlands, particularly vulnerable isolated wetlands, on streamflow. As such, this work will likely be a substantial contribution to both the hydrologic sciences and national to global policies related to wetland conservation and mitigation. The narrative and figures are concise and effectively convey the key, and again important, contributions; with that said, I offer some suggested areas for more clarification/expansion below in detailed comments. But, in short, and given the scope and impact of this work, I recommend that it is well suited for Communications Earth and Environment.

Detailed Comments:

Abstract:

As I mention in later comments, I suggest adding more about the approach that was used to understand differences in catchment-level patterns (i.e., runoff vs. MIWA) and more on the findings. That is, the work to identify catchment differences (linear vs. threshold) and attempt to understand the potential reasons why (GIWs vs. other catchment properties) could be elevated.

Also, I suggest adding a bit of text regarding the statistical approach to separate the two focal drivers; without that, readers may question the work as clearly these covary.

Introduction

Lines 70-72: Consider flipping the order to reflect the process...fill before spill. Also, "in-situ sequestration *and removal*".

Line 73: I'd be cautious to say "limited number", and it's not really needed here.

Line 78: Depending on the audience, runoff can mean different things. I suggest defining as it's used here explicitly.

Line 80: Here and in many other places, "reference" is used vs. the actual reference.

Line 82: "broader climate seasonality" is vague here, especially given the subject is already "intra-annual climate variability", thus the same thing.

Line 83: Include topography?

Line 84: replace "slightly" with *can*.

Lines 87-95: Related to my comment re: abstract, this would be a good place to emphasize how catchments may exhibit different behaviors and the related knowledge gaps. As written, this paragraph and text in the following paragraph is more about knowledge gaps re: the general role of MIWA vs. climate and if the response is a threshold one, failing to address that catchments may vary in the relative roles of MIWA/climate and in their runoff response. A few added sentences could go a long way here to set the stage of the catchment-specific analyses to come.

Line 99: “drainage continuous to reduce *wetland water storage capacity and overall wetland extent*” Suggested change to reflect that wetland area can still remain but with reduced storage capacity.

Lines 103-107: This state re: contribution of the work breaks up flow between the preceding state that sets up the gap (It remains uncertain) and the following text to address the gap (To address this knowledge gap).

Line 127: “isolate depressional wetland inundation”. If I’m understanding it correctly, only the river/stream and lake features were removed, not their floodplains/riparian areas, as the latter were used in MIWA, correct? This is the correct approach in my opinion: to consider both riparian and GIWs, and then how they modulate the RO vs. MIWA response differently. But, the use of depressional wetland inundation here may suggest to some (including me) that riparian (which may not be depressional) were not included in this analysis and were removed with the streams. So, maybe a simple edit (provided that I’m correct) to say that rivers and lakes were removed to isolate wetland areas, furthering characterizing them as either riparian or GIWs.

Line 128: Annual MIWA. Was the time series of MIWA examined for wetland loss trends? I see later, which I appreciate, that this analysis allowed this to be implicitly included (i.e., the effect of lower MIWA values, regardless of why) and thus to suggest the role of wetland drainage on runoff. But, and maybe not here, could there be more attention to any insights for wetland loss using the time series of MIWA, and then what that meant for runoff?

Lines 150-153: Seems to repeat points from before.

Line 159: While I understand the use of “diminishing” for saturating, its connotation is misleading a bit here, suggesting the MIWA has a diminishing effect on runoff. Rather, more increases in MIWA means increasing storage effects, correct? Consider rephrasing to emphasize that.

Lines 163-169: You also assessed other drivers of the exponent b ; I suggest including that here. Indeed, that was somewhat lost on me until I got to the methods, as it was only briefly mentioned in the results. There are likely opportunities to sprinkle in that you also assessed other drivers more explicitly.

Line 164: When I first read this, I thought that the explanatory variable was percent GIW vs. percent riparian area (not time-varying inundated area), thus a static variable for catchments unless drainage reduced wetland area. But from the methods, I see that it was the maximum inundation for each of these categories over the 38 years. That’s fine, but that could be clarified here.

Lines 239-240: Consider just providing the percentages of each case.

Lines 241-243: Consider bringing up this case-specific example (that leans on Fig 4a) to start this section to then transition to your approach used across all catchments to systematically assess and quantify the different possible patterns (functional forms).

Line 263: I believe this should be: "...responsive, *linear* regime to one dominated..." since the $b = 1.06$.

Line 270: Consider a word other than "resistant", as MIWA is not resistant to flows but decreases them until...

Line 284: The role of GIWs in regulating hydrologic connectivity is not episodic, as their presence is always regulating the degree of connectivity...their SW connectivity is just episodic. Revise.

Lines 297-299: Well said. Indeed, and as I mentioned before, your approach allows an assessment of how changes in MIWA (regardless of reason) affect runoff. That could maybe be emphasized here and earlier.

Line 331: It's not just 'even when' but because of their disconnection and only episodic connections. That could be emphasized more.

Methods:

Line 147: How was AET derived?

Lines 514-515: Same comment as before that this ("depressional wetlands and potholes") may sound to some that you excluded riparian wetlands.

Lines 537-543: This is results that could lead off the results section to then reference Fig 1b, c, which are main body figures but not referenced in the main body.